# Towards Evaluating Transfer-based Attacks Systematically, Practically, and Fairly

**Qizhang Li**[1,2], **Yiwen Guo**[3*], **Wangmeng Zuo**[1], **Hao Chen**[4]
[1]Harbin Institute of Technology, [2]Tencent Security Big Data Lab, [3]Independent Researcher, [4]UC Davis
{liqizhang95,guoyiwen89}@gmail.com  wmzuo@hit.edu.cn  chen@ucdavis.edu

## Abstract

The adversarial vulnerability of deep neural networks (DNNs) has drawn great attention due to the security risk of applying these models in real-world applications. Based on transferability of adversarial examples, an increasing number of transfer-based methods have been developed to fool black-box DNN models whose architecture and parameters are inaccessible. Although tremendous effort has been exerted, there still lacks a standardized benchmark that could be taken advantage of to compare these methods systematically, fairly, and practically. Our investigation shows that the evaluation of some methods needs to be more reasonable and more thorough to verify their effectiveness, to avoid, for example, unfair comparison and insufficient consideration of possible substitute/victim models. Therefore, we establish a transfer-based attack benchmark (TA-Bench) which implements 30+ methods. In this paper, we evaluate and compare them comprehensively on 25 popular substitute/victim models on ImageNet. New insights about the effectiveness of these methods are gained and guidelines for future evaluations are provided. Code at: https://github.com/qizhangli/TA-Bench.

## 1  Introduction

In recent years, deep neural networks (DNNs) have demonstrated unprecedented success in various applications. However, the success comes at a price: DNNs are vulnerable to adversarial examples crafted by adding imperceptible perturbations to inputs (*e.g.*, images). The existence of adversarial examples poses a significant threat to the security and reliability of DNNs, especially in safety-critical applications such as autonomous driving, biometrics, and medical image analysis.

There are many different ways of generating adversarial examples and performing attacks. Transfer-based attacks, that are capability of compromising DNNs without having access to their network architecture and parameters, have been widely studied over the past few years. To issue such transfer-based attacks, an attacker first collects a substitute model or a set of substitute models, then computes gradients on the substitute model(s) and perform optimization based on the gradients. Performance of the attacks largely rely on the transferability of the generated adversarial examples. Over the years, a large number of methods have been proposed to improve the adversarial transferability. The methods innovate various aspects of the attack procedure, ranging from *substitute model training* [47, 78, 13, 27] to *gradient computation* (that modifies loss or forward/backward architectures given well-trained substitute models) [77, 11, 20, 14, 65, 26, 60, 57, 61, 74, 21, 15, 74] to *input augmentation* [67, 7, 28, 58] *and the optimizer* that applies the gradients [6, 28, 59]. Some methods further propose to train a generative model using additional data for obtaining transferable adversarial examples [35, 44, 75, 36, 69].

---

*Yiwen Guo leads the project and serves as the corresponding author.

37th Conference on Neural Information Processing Systems (NeurIPS 2023).

Despite all the progress, there still lacks a standardized benchmark that could be taken advantage of to compare these methods systematically, fairly, and practically. Existing evaluations in the literature suffer from several limitations. First, there is a paucity of variety in the tested models, while certain methods may exhibit superior performance only with specific substitute/victim architectures. Second, many methods verified their efficacy only with a basic optimization back-end, *e.g.*, I-FGSM [24], despite the existence of more advanced optimization including input augmentations. It is hence unclear whether the benefits of an innovation in these methods can be similarly brought by incorporating these more advanced optimization techniques, which could undermine the reliability of the evaluations.

To address these problems, we create a transfer-based attack benchmark (TA-Bench), allowing researchers to compare a variety of methods in a fair and reliable manner. We believe that TA-Bench will foster the development of adversarial machine learning and inspire effective ways of generating adversarial examples for evaluating the robustness of DNNs. Our main contributions include (but not limited to) the following items.

**A More Advanced Optimization Back-end.** We have evaluated various combinations of input augmentation mechanisms and optimizers on our TA-Bench. Our results convey that, with a few exceptions, combining more types of augmentations leads to more powerful attacks, and it could be more reasonable to evaluate the effectiveness of input augmentation mechanisms and optimizers in combination rather than in isolation. Besides, the obtained combination is suggested to be a more advanced optimization back-end for evaluating the "gradient computation" methods and "substitute model training" methods.

**Insightful Observations from Comprehensive Evaluations.** We consider various substitute/victim models that are considered popular, including CNNs (*e.g.*, ResNet-50 [16], VGG-19 [49] with batch normalization, Inception v3 [49], EfficientNetV2-M [51], and ConvNeXt-B [32]), vision transformers (*e.g.*, ViT-B [8], DeiT-B [54], Swin-B [31], and BEiT-B [1]), and a MLP (MLP-Mixer [53]). Comprehensive results on our TA-Bench systematically shows how the performance of transfer-based attacks varies with different choices of substitute/victim models. These results demonstrate that adopting transformers as the substitute model generally yields superior attack performance for "gradient computation" methods, comparing with using traditional convolutional models (as the substitute model).

**A Unified Codebase.** We offer an open-source codebase for TA-Bench, featuring a well-organized code structure that can effectively accommodate a diverse range of transfer-based attacks, as well as various substitute/victim models. It provides a unified setting for evaluations, ensuring consistency and reproducibility in experimental results. The code is at https://github.com/qizhangli/TA-Bench.

## 2   Related Work

**Transfer-based Attacks.**  Transfer-based attacks emerged as a result of the discovery that adversarial examples are not only powerful against the model they were generated on but also effective against other models. By exploiting such a discovery, attackers can use adversarial examples created on some substitute models to attack the victim model, and a series of transfer-based attacks have been proposed. These methods have innovated various aspects of the attack procedure, and we will carefully discuss them in Section 3.2.

**Related Benchmarks.** There have been several libraries or benchmarks for generating adversarial examples, such as CleverHans [39], Foolbox [41, 42], RobustART [52], Torchattacks [23], *etc.* However, these benchmarks only cover a very limited number of transfer-based methods, as they were developed for evaluating the robustness of DNNs against not only black-box attacks but also white-box attacks. In particular, most of the methods are outdated and only considered as baseline methods due to the rapid development in this field. By contrast, our TA-Bench focuses on transfer-based attacks, and it implements 30+ methods in order to perform systematical, practical, and fair comparisons. A contemporary benchmark [76] also evaluates many transfer-based attacks and it draws attention to the stealthiness of the adversarial examples. Our contributions are mostly orthogonal to theirs, as we focus on fair and practical comparison between different methods. We will show some previous conclusions might be overthrown in practice. We provide new insights and guidance for evaluation to assist the future work in the community.

# 3 Our Benchmark

## 3.1 Threat Model

In a black-box scenario, the adversary has limited access to the victim model. Transfer-based attack aims to compromise the victim model by generating adversarial examples on a substitute model or a set of substitute models. Recent transfer-based methods, with few exceptions [25, 48], tested in a setting where: 1) training data of the victim model is accessible and training/fine-tuning a model on such data is possible for the adversary [47, 78, 13, 27, 40, 35, 44, 75, 36], or 2) the adversary is able to collect at least one substitute model trained using the same dataset that the victim model learned from [37, 77, 22, 11, 20, 14, 65, 26, 60, 57, 61, 74, 21, 15, 74, 24, 34, 67, 7, 58, 6, 28]. We follow this assumption for setting up the benchmark. Specifically, nothing except for the training data is known about the victim model for performing attacks in our threat model, *i.e.*, the adversary has no idea about the pre-processing pipeline, architecture, and parameters.

Adversarial examples on the benchmark are all obtained by performing pixel-wise perturbations to benign images under an $\ell_p$ constraint.

## 3.2 Methodologies

In order to compare different methods more reasonably, we roughly divide existing methods, based on their main innovations, into four categories highlighted as follows.

**Input Augmentation and Optimizer.** To craft an adversarial example on any given substitute model, the perturbation can be optimized as in the white-box setting. Gradient-based iterative optimization is commonly utilized, in which the perturbation is initialized to a zero tensor (*e.g.*, in I-FGSM [24]) or a random tensor whose entries are sampled from a distribution (*e.g.*, in PGD [34]). Image data augmentation has been considered for generating transformation-robust perturbations to each benign example, for example in diverse inputs I-FGSM (DI$^2$-FGSM) [67], translation-invariant I-FGSM (TI-FGSM) [7], scale-invariant I-FGSM (SI-FGSM) [28], and Admix [58]. Several attacks also innovate by taking advantage of the momentum optimizer, *e.g.*, momentum I-FGSM (MI-FGSM) [6], Nesterov I-FGSM (NI-FGSM) [28], and pre-gradient guided momentum I-FGSM (PI-FGSM) [59]. In general, these methods are all architecture-independent.

**Gradient Computation (DNN-Specific).** There is a belief that improved adversarial transferability can be achieved by modifying the loss or the backpropagation process. For backpropagation, both the forward and the backward pass can be altered to achieve powerful attacks, and a series of methods, including SGM [65], LinBP [14], ConBP [72], PNA [62], and SE [38] have been proposed. Some methods advocate loss terms obtained on a middle layer of the substitute model (*e.g.*, NRDM [37], TAP [77], FDA [11], ILA [20], ILA++ [26], FIA [61], NAA [74]), while other methods stick with loss computed on the final layers (*e.g.*, IR [60], VT [57], and TAIG [21]).

**Substitute Model Training.** Though most prior work uses off-the-shelf models (that could be collected on the Internet) directly as substitute models, some methods advocate fine-tuning these models or even training new ones to better suit the goal of achieving transferable adversarial examples. For instance, RFA [47] suggests adopting adversarial training to obtain substitute models. DRA [78] fine-tunes the substitute model to push the adversarial examples away from the distribution of their benign counterparts during performing attacks. LGV [13] fine-tunes the substitute model with a high learning rate and to collect a set of models on the training trajectory. A very recent method proposed by Li *et al.* performs fine-tuning in order to obtain Bayesian substitute models [27].

**Generative Modeling.** In addition to the above mentioned attacks, there are another line of transfer-based attacks that use generative models to craft adversarial examples. These methods often require additional data for training the generative model. Once properly trained, the model can generate transferable adversarial examples across different victim models. Some effective methods concerned in this line include CDA [35], GAPF [44], BIA [75], TTP [36], C-GSP [69], *etc.*

**All these mentioned methods have been implemented on our benchmark.** That is, we implemented **30+ methods** from these four categories for comprehensive evaluation and comparison of transfer-based attacks.

## 3.3 Victim Models and Substitute Models

With a surge of innovation in deep learning, there exists a variety of image classification DNNs. Each has its own advantages. In practice, an engineer can choose any of them to train and deploy according to his or her specific requests. An adversary whose aim is to compromise a computer vision service developed by the engineer then anticipate the generated adversarial examples transfer well to all these possible models. Hence, to make the evaluation comprehensive and practical, we consider various victim models that are considered popular, including **CNNs** (*e.g.*, ResNet-50 [16], VGG-19 [49] with batch normalization, Inception v3 [49], EfficientNetV2-M [51], and ConvNeXt-B [32]), **vision transformers** (*e.g.*, ViT-B [8], DeiT-B [54], Swin-B [31], and BEiT-B [1]), and **a MLP** (MLP-Mixer-B [53]). It is worth noting that all of these models were obtained directly from an open-source repository `timm` [63] on GitHub, and our benchmark can be easily updated to include new models in the future.

Having witnessed the development of image classification architectures, it is unwise for the adversary to stick with conventional models (*e.g.*, ResNets, VGG-Nets, and Inceptions), since it is possible that generating adversarial examples on a more advanced substitute model leads to superior attack success rates. Thus, on this benchmark, we employ the 10 victim models named above to generate adversarial examples and evaluate the attack performance of all options. As will be shown in Section 4.3, many new insights can be gained from the evaluation results.

## 3.4 Experimental Settings and Implementation

All evaluations on our benchmark are conducted on ImageNet [43]. We randomly selected 5,000 benign examples that could be correctly classified by all the victim models, from the ImageNet validation set, to craft adversarial examples. Filenames of these benign examples will be provided, for reproducing results and testing new methods in the future. A distance metric is required to measure the magnitude of perturbations. We adopt the popular $\ell_p$ distance for $p \in \{\infty, 2\}$ and set the perturbation budget under $\ell_\infty$ and $\ell_2$ constraints to $\epsilon = 8/255$ and $\epsilon = 5$, respectively, to guarantee that the adversarial perturbations are almost imperceptible. The optimization process of each compared method runs $100$ iterations with a step size of $1/255$ and $1$ for $\ell_\infty$ constraint and $\ell_2$ constraint, respectively. For each victim model, we pre-process its input in exactly the same way as their official implementation to ensure a practical evaluation of attack performance, and note that the adversary has no idea about the detailed implementation of this pre-processing. For instance, when evaluating the performance of attacking a ResNet-50 victim, we first resize an adversarial example to $256 \times 256$ by bilinear interpolation, then crop the image to $224 \times 224$ in the center, and finally feed the $224 \times 224$ image into the victim model and evaluate whether the attack is successful or not. Implementation details about all the supported methods are provided in Section F in the Appendix. All experiments are performed on an NVIDIA V100 GPU.

For evaluating the transferability of adversarial examples, we use the accuracy of victim models for classifying the adversarial examples as a measure. Using the prediction accuracy of victim models instead of the attack success rate makes it easier to incorporate other victim models in the future, as a reasonable calculation of attack success rate often requires the benign counterparts of the adversarial examples be classified correctly by all victim models. With a specific choice of the substitute model, prior work often evaluates the average accuracy (AA) over all victim models for comparing different attacks. However, since our benchmark studies a variety of substitute models, we further evaluate the average AA (AAA), the worst AA (WAA), and the best AA (BAA) over all choices of these substitute models. **Lower AAA, WAA, and BAA indicate stronger attacks and more vulnerable victim models.**

We have built a codebase consisting of modular components that serve as the basis of TA-Bench. By leveraging modular design principles, the substitute and victim models, back-end methods, and hyper-parameters can be easily adapted to help the future work of the community.

# 4 Evaluations and Analyses

In Section 4.1, we identify a pre-processing pipeline that is more practical. In Section 4.2, we investigate an improved back-end method by evaluating the possible combinations of iterative optimization methods. We then re-evaluate state-of-the-arts under a comprehensive and unified

benchmark, which incorporates various substitute and victim models, to assess the effectiveness of these methods in Section 4.3 and Section 4.4.

## 4.1 Pre-processing in Practice

Modern image classification models (especially CNNs) can take images of various sizes. After investigating experimental settings of previous transfer-based attacks, we found that the adversarial examples were often evaluated by feeding them into the victim models directly (without taking pre-processing operations of the victim models into account, *e.g.*, resize and crop) [28, 57, 59, 58] or just resize to the input size of the victim models [61, 74]. However, these victim models, when deployed, are often equipped with inference time data pre-processing to improve their effectiveness and efficiency. Getting rid of it may lead to over-estimation of adversarial vulnerability.

As mentioned in Section 3.4, on our benchmark, we follow the default pre-processing pipeline of each victim model and evaluate under the circumstances where pre-processing often exists. We observed degraded attack performance under such circumstances (see Figure 1 for results). Obviously, the adversarial examples seem less vulnerable when the default test-time pre-processing is re-introduced to each victim model. Given the same I-FGSM adversarial examples, the average accuracy of victim models increases to 87.79% (from 86.16%) with pre-processing, confirming that **ignoring pre-processing operations of victim models indeed leads to over-estimation of their adversarial vulnerability**. Similar observations can be made with other substitute models and other attack methods. Considering that the pre-processing pipeline of the victim model is inaccessible to the adversary, it is infeasible for the adversary to follow the same

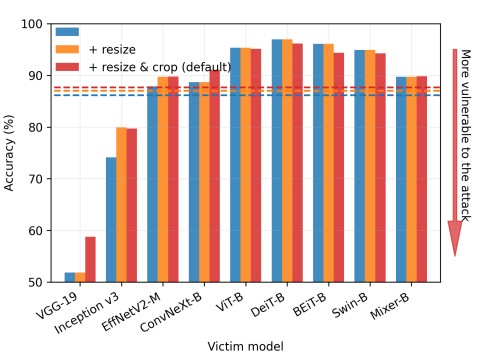

Figure 1: Attack performance with different pre-processing strategies in the target environment, and the AAA is indicated by dotted lines (lower means more vulnerable to the attack). The adversarial examples were generated using I-FGSM on a ResNet-50 substitute model under the $\ell_\infty$ constraint with $\epsilon = 8/255$.

pipeline when generating adversarial examples. On the substitute models, a safe choice is to resize each of their inputs to the default size without cropping them.

## 4.2 Evaluation of Input Augmentation and the Optimizer

To verifying the effectiveness of a newly developed computer vision architecture (*e.g.*, vision transformers), it is common to show that its performance surpasses previous state-of-the-arts on a challenging benchmark dataset (*e.g.*, ImageNet), and test of the newly developed architecture may utilize a combination of advanced optimization strategies (*e.g.*, AdamW [33] + mixup [73] + cutmix [71] + stochastic depth [19]) if such a combination is beneficial [2, 55]. Novel optimization strategies are also often tested in combination with existing ones, to show their consistent effectiveness [12].

Likewise, the transferability of adversarial examples can also benefit from appropriate choices of input augmentations and the optimizer, however, ways of innovating these optimization strategies are often evaluated in isolation in this setting. On our benchmark, we, for the first time, evaluate combinations of different choices of input augmentations and the optimizer systematically. Specifically, we performed a grid search to seek the optimal combination of I-FGSM [24], PGD [34], DI$^2$-FGSM [67], TI-FGSM [7], SI-FGSM [28], Admix [58], NI-FGSM [28], MI-FGSM [6], PI-FGSM [59], *etc*. Since Admix inherently includes SI-FGSM, we have SI-FGSM by default when Admix is chosen. NI-FGSM, MI-FGSM, and PI-FGSM seem not orthogonal and thus they are tested in a mutually exclusive way in our experiment. The same for I-FGSM and PGD. For TI-FGSM, an alternative implementation where inputs are translated is adopted, as it is more effective than its suggested approximation which convolves gradients. Besides the augmentations in DI$^2$-FGSM, TI-FGSM, SI-FGSM, and Admix, we consider several other input augmentation mechanisms including adding uniform noise to the input (which was used in VT [57] and TAIG [21]) and randomly dropping some patches of the perturbation (which was used in IR [60]). We call the two methods as UN and DP, respectively. Note that in a previous work [66], Gaussian noise is added to the input in each attack iteration. In this study, we

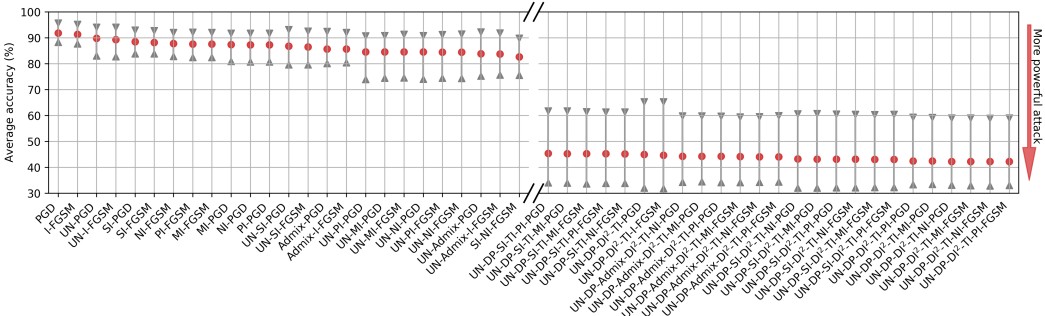

Figure 2: Comparing different combinations of the optimization strategies. The red solid circles indicate AAA, while the grey triangles show BAA and WAA (lower indicates more powerful attack).

opt for uniform noise, as it is now more commonly adopted. It is also worth noting that the original implementation of SI-FGSM and Admix uses several augmented copies of an input and averages the gradients computed on these copies for optimization. Such an approach increases the computational complexity of performing attacks, and, in fact, all input augmentation mechanisms in this category can be improved by such an approach [76], and, for reducing computational complexity, we only craft one augmented input at each iteration.

Each possible combination is tested with every possible substitute model whose name has been mentioned in Section 3.3 to evaluate the attacking performance over other models which are considered as the victim models. The results are sorted by AAA, in a descending order from left to right, and demonstrated in Figure 2. It illustrates not only AAA but also the range between BAA and WAA (as "error bars"). **We see that the optimal AAA is achieved by UN-DP-DI$^2$-TI-PI-FGSM, which is 42.42%.** The detailed AA when each substitute model is chosen for crafting adversarial examples by the method is reported in Section E. Inspecting the obtained results, we found that the performance gap between MI-FGSM, NI-FGSM, and PI-FGSM is marginal. For instance, the top 3 solutions (with PI-FGSM, NI-FGSM, and MI-FGSM, respectively) lead to similar AAA (42.42%, 42.45%, and 42.46%). The performance of I-FGSM and PGD is also similar. In most cases, PGD leads to slightly inferior performance than that of I-FGSM, accordingly to our experimental results.

In general, more input augmentation mechanisms leads to more powerful attacks. Yet there are exceptions. With DP adopted, SI-FGSM and Admix that apply input scaling and mixing fail to manifest their gains regarding the adversarial transferability. In particular, the AAA increases to 43.12% and 44.11% when further adding SI and Admix to UN-DP-DI$^2$-TI-PI-FGSM, respectively. This is in contrast to the previous belief that these two methods are effective, and a possible explanation is that, when adopting DP and Admix/SI-FGSM simultaneously, the augmentation becomes too strong to keep the input a in-distribution sample. We adopted the default hyper-parameters for all combined methods and it is possible (yet computationally very intensive since the number of combinations is huge) to carefully tune hyper-parameters to achieve even better combinations. We will leave it to future work.

### 4.3 Evaluation of "Gradient Computation" Methods and "Substitute Model Training" Methods

The previous section evaluates input augmentations and optimizers, and, in this section, we shall focus on "gradient computation" and "substitute model training" methods.

Firstly, we would like to emphasize that certain gradient computation innovations (*e.g.*, IR [60], VT [57], and TAIG [21]) incorporate input augmentations for averaging the gradients obtained from multiple randomly augmented inputs at each iteration. These methods require high computational cost, and comparing them with other methods in the same category directly is unfair, as the other methods can also apply random augmentation and gradient averaging to boost their performance. Taking VT [57] as an example, it introduces random noise to the input and perform backpropagation for 20 times at each iteration. Similar for IR and TAIG. We compare them with their corresponding baselines that employ the same input augmentations and keep the same number of backpropagation operations at each iteration. That is, the corresponding baselines of VT and IR (*i.e.*, VT-baseline and IR-baseline) perform multiple rounds of backpropagation with UN and DP augmentations, respectively, at each

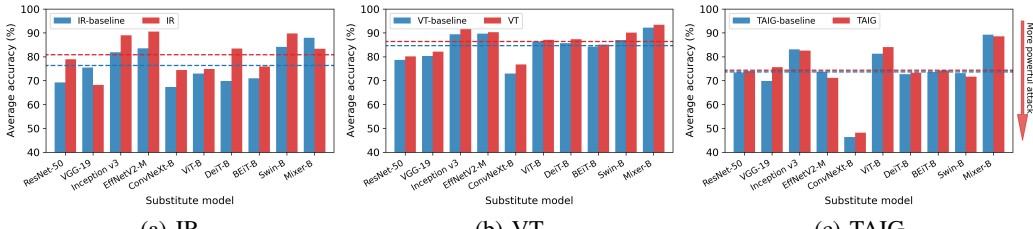

Figure 3: Comparing IR, VT, and TAIG with their corresponding baselines. The dotted lines indicate AAA (lower indicates more powerful attack).

iteration of optimization, and the corresponding baseline of TAIG (*i.e.*, TAIG-baseline) perform both input scaling and UN augmentations. Figure 3 reports the comparison results. It can be seen that the simple baselines achieve even lower AAA, **indicating that the most effective factor of these methods may be input augmentation and gradient averaging, instead of what were claimed.** Based on these findings, we strongly recommend that newly designed methods, which incorporate input augmentations, either explicitly or implicitly, should be carefully compared with reasonable baselines that adopt the same mechanisms to show their effectiveness.

There are still 10+ methods to be evaluated and compared. Previous work often compares using a simple optimization back-end, *e.g.*, I-FGSM or MI-FGSM. As we have obtained a combination of optimization strategies (*i.e.*, UN-DP-DI$^2$-TI-PI-FGSM) which has been proven to be more powerful in Section 4.2, we further introduce it as a more advanced optimization back-end. It aids in better exploring true advantages of compared methods. To ensure optimal performance across different substitute models, we employed a validation set consisting of 500 samples that were distinct from the test examples tune hyper-parameters of compared methods. The hyper-parameters that yielded the best results on the validation set were then adopted for testing. The hyper-parameters with each substitute model are tuned using models whose name have been mentioned in Section 3.3. The attack performance is evaluated not only on these "validation" models, but also on 15 more victim models which are distinct from them. The detailed hyper-parameters are reported in Section F.

In Table 1 (column 2-11), we report the AA achieved by utilizing each model as the substitute model to attack the 9 "validation" models under the $\ell_\infty$ constraint. AAA (which is the average AA over all substitute models) is provided in the 12-th column in the table. Note that some methods are unable to be performed on some architectures. For instance, ConBP [72] suggests replacing ReLU with a softplus function in the backward pass to ensure smooth gradient backpropagation, making it not suitable to substitute models that are equipped with smooth activations only. SGM [65] is not applicable to substitute architectures without skip-connections (*e.g.*, VGG-19). PNA [62] and SE [38] focus on vision transformers, and thus they are not suitable to most convolutional substitute models. For "substitute model training" methods, ResNet-50 is commonly chosen as the substitute model, as is adopted in the official GitHub implementations of these methods, and we leave the exploration of training with more advanced substitute architectures to future work. In addition to the $\ell_\infty$ experiment, we also conduct evaluate with the $\ell_2$ constraint and the results are provided in Section A.

When I-FGSM [24] is simply applied as the optimization back-end, NAA [74] beats the other gradient computation innovations and achieves the lowest AAA with a value of 72.65% (see the upper half of Table 1), while RFA demonstrates the best results among all "substitute model training" methods. Table 1 shows that the most suitable gradient computation strategy can be different for issuing attacks from different substitute models. In particular, if ResNet-50 or VGG-19 is chosen as the substitute model, FIA [61] seems even superior to NAA in the sense of achiving higher attack success rate and lower victim accuracy, however, NAA is the best if any other model is chosen as the substitute model.

When UN-DP-DI$^2$-TI-PI-FGSM is introduced as the new optimization back-end, almost all "gradient computation" methods show improved performance (see the lower half of Table 1). Yet, in combination with the new baseline, the advantage of most methods becomes less obvious. SGM [65], PNA [62], and SE [38] still produce performance gains when being combined with UN-DP-DI$^2$-TI-PI-FGSM. In particular, with the new optimization back-end, SE on the DeiT-B substitute model leads to the optimal attack performance among all concerned options, in the sense of BAA, fooling the victim models to show an accuracy of only 18.61% (on average). PNA obtains the lowest WAA among all, which is 31.50%. As for substitute model training, we see that the MoreBayesian method significantly outperforms the other methods, and it fools the victim models to show an average accuracy of 27.09% using a ResNet-50 substitute model.

Table 1: Comparing the obtained AA and AAA of some "gradient computation" and "substitute model training" methods. Smaller values indicate more powerful attacks. The adversarial examples were generated under an $\ell_\infty$ constraint with $\epsilon = 8/255$.

| | ResNet-50 | VGG-19 | Inception v3 | EffNetV2-M | ConvNeXt-B | ViT-B | DeiT-B | BEiT-B | Swin-B | Mixer-B | AAA |
|---|---|---|---|---|---|---|---|---|---|---|---|
| **I-FGSM Back-end** | | | | | | | | | | | |
| **- Baseline** | | | | | | | | | | | |
| I-FGSM | 87.79% | 91.21% | 93.71% | 95.46% | 88.32% | 90.28% | 90.28% | 89.56% | 94.81% | 94.37% | 91.58% |
| **- Gradient Computation** | | | | | | | | | | | |
| TAP (2018) [77] | 81.75% | 89.80% | 91.01% | 93.84% | 90.20% | 91.90% | 92.86% | 92.11% | 95.08% | 93.93% | 91.25% |
| NRDM (2018) [37] | 82.19% | 87.62% | 85.29% | 96.12% | 94.36% | 94.70% | 95.02% | 95.23% | 95.01% | 90.25% | 91.58% |
| FDA (2019) [11] | 85.11% | 93.91% | 89.91% | 98.00% | 96.27% | 96.60% | 95.52% | 96.67% | 97.56% | 97.66% | 94.72% |
| ILA (2019) [20] | 74.76% | 77.21% | 83.38% | 90.20% | 84.13% | 77.91% | 80.62% | 78.29% | 89.18% | 85.30% | 82.10% |
| SGM (2020) [65] | 72.56% | - | - | 79.64% | 71.37% | 85.72% | 87.04% | 83.67% | 90.55% | 91.01% | - |
| ILA++ (2020) [26] | 71.80% | 73.60% | 80.07% | 88.01% | 83.12% | 74.50% | 80.19% | 77.02% | 88.08% | 82.08% | 79.85% |
| LinBP (2020) [14] | 75.84% | 86.66% | 92.87% | 96.96% | 89.05% | 91.74% | 91.26% | 92.62% | 95.65% | 96.07% | 90.87% |
| ConBP (2021) [72] | 73.46% | 85.49% | 91.00% | - | - | - | - | - | - | - | - |
| SE (2021) [38] | - | - | - | - | - | 90.15% | 89.18% | 89.34% | - | 92.34% | - |
| FIA (2021) [61] | 68.48% | 71.86% | 83.84% | 89.66% | 80.35% | 76.06% | 80.13% | 82.42% | 88.75% | 79.13% | 80.07% |
| PNA (2022) [62] | - | - | - | - | - | 88.13% | 87.14% | 87.97% | 93.62% | - | - |
| NAA (2022) [74] | 70.34% | 78.41% | 76.37% | 83.56% | 63.93% | 65.04% | 69.02% | 66.24% | 79.26% | 74.33% | 72.65% |
| **- Substitute Model Training** | | | | | | | | | | | |
| RFA (2021) [47] | **47.49%** | - | - | - | - | - | - | - | - | - | - |
| LGV (2022) [13] | 74.84% | - | - | - | - | - | - | - | - | - | - |
| DRA (2022) [78] | 48.55% | - | - | - | - | - | - | - | - | - | - |
| MoreBayesian (2023) [27] | 63.40% | - | - | - | - | - | - | - | - | - | - |
| **New Optimization Back-end** | | | | | | | | | | | |
| **- Baseline** | | | | | | | | | | | |
| UN-DP-DI$^2$-TI-PI-FGSM | 35.70% | 48.33% | 58.62% | 52.98% | 33.64% | 32.74% | 36.58% | 33.72% | 45.24% | 46.60% | 42.42% |
| **- Gradient Computation** | | | | | | | | | | | |
| TAP (2018) [77] | 63.34% | 54.64% | 68.02% | 68.90% | 27.26% | 41.48% | 46.78% | 34.45% | 56.02% | 54.49% | 51.54% |
| NRDM (2018) [37] | 51.78% | 63.14% | 70.76% | 61.81% | 40.04% | 52.12% | 60.89% | 48.98% | 77.87% | 57.84% | 58.52% |
| FDA (2019) [11] | 42.62% | 52.83% | 60.25% | 89.48% | 69.01% | 94.83% | 83.99% | 78.26% | 83.19% | 94.97% | 74.94% |
| ILA (2019) [20] | 37.80% | 45.66% | 54.99% | 48.86% | 30.72% | 28.98% | 33.28% | 29.40% | 56.98% | 45.80% | **41.25%** |
| SGM (2020) [65] | **31.97%** | - | - | **27.82%** | 20.96% | **28.80%** | 24.77% | 25.42% | **24.27%** | 38.57% | - |
| ILA++ (2020) [26] | 37.26% | 44.77% | **54.24%** | 48.62% | 31.15% | 29.49% | 34.68% | 29.80% | 59.30% | 45.66% | 41.50% |
| LinBP (2020) [14] | 38.00% | 48.18% | 80.28% | 90.10% | **18.81%** | 41.44% | 36.89% | 45.51% | 70.28% | 82.75% | 55.22% |
| ConBP (2021) [72] | 36.10% | 48.15% | 69.73% | - | - | - | - | - | - | - | - |
| SE (2021) [38] | - | - | - | - | - | 35.93% | **18.61%** | 24.70% | - | **35.58%** | - |
| FIA (2021) [61] | 38.22% | 52.38% | 58.38% | 76.93% | 51.21% | 42.31% | 42.28% | 59.02% | 62.00% | 58.65% | 54.14% |
| PNA (2022) [62] | - | - | - | - | - | 31.50% | 19.76% | 28.88% | 29.80% | - | - |
| NAA (2022) [74] | 38.04% | 49.58% | 54.66% | 54.42% | 32.04% | 31.45% | 33.34% | 41.10% | 50.07% | 41.28% | 42.60% |
| **- Substitute Model Training** | | | | | | | | | | | |
| RFA (2021) [47] | 43.00% | - | - | - | - | - | - | - | - | - | - |
| LGV (2022) [13] | 31.82% | - | - | - | - | - | - | - | - | - | - |
| DRA (2022) [78] | 51.10% | - | - | - | - | - | - | - | - | - | - |
| MoreBayesian (2023) [27] | **27.09%** | - | - | - | - | - | - | - | - | - | - |

According to Table 1, vision transformers should be preferable when choosing the substitute model, as the best attack performance (*i.e.*, BAA) is often obtained on vision transformers for many attacks. To compare the transfer performance from vision transformers to convolutional networks and from the opposite direction, we report the accuracy of victim models in predicting SGM adversarial examples generated on ResNet-50 and ViT-B as the substitute model, respectively. The results are shown in Table 2. It can be seen that transferring from vision transformers to convolutional networks seems easier. When utilizing ViT-B as the substitute model, the accuracy of convolutional networks shows a range in $[28.32\%, 37.24\%]$, while, with ResNet-50, the accuracy of vision transformers lies in $[36.82\%, 48.32\%]$. Overall, using ViT-B as the substitute model leads to lower average accuracy ($28.80\%$ vs $31.97\%$) and the worst accuracy ($37.24\%$ vs $48.32\%$) on victim models, which means better average and worst-case attack performance, respectively. ConvNeXt [32] that follows some designing principles of the vision transformers is also a favorable choice of the substitute model, according to our results. When performing LinBP on ConvNeXt-B, the generated adversarial examples are capable of fooling the victim models to show an average accuracy of only $18.81\%$, which is super close to the best attack performance that could be achieved in Table 1. Additionally, it

Table 2: The accuracy of victim models in predicting adversarial examples crafted via SGM using ResNet-50 and ViT-B as the substitute model, respectively. Smaller values indicate more powerful attacks. The optimization back-end is UN-DP-DI$^2$-TI-PI-FGSM, and the adversarial examples were generated under an $\ell_\infty$ constraint with $\epsilon = 8/255$.

| Substitute model | ResNet-50 | VGG-19 | Inception v3 | EffNetV2-M | ConvNeXt-B | ViT-B | DeiT-B | BEiT-B | Swin-B | Mixer-B | AA |
|---|---|---|---|---|---|---|---|---|---|---|---|
| ResNet-50 | - | 2.72% | 7.92% | 29.42% | 28.52% | 48.32% | 47.64% | 36.82% | 47.66% | 38.70% | 31.97% |
| ViT-B | 30.00% | 28.32% | 36.40% | 37.24% | 33.66% | - | 28.76% | 15.60% | 23.26% | 25.92% | 28.80% |

is worth noting that even though ViT-B, DeiT-B, and BEiT-B share the same network architecture, they exhibit considerably different performance when acting as the substitute model. This suggests that the training procedure can also play a crucial role in improving the adversarial transferability.

If we still focus on ResNet/Inception and test adversarial examples only on these traditional models, as in many previous papers, then different conclusions will be drawn. Likewise, if we only focus on the simple optimization back-end (*i.e.*, I-FGSM) without introducing UN-DP-DI$^2$-TI-PI-FGSM, the conclusions will also be different, since NAA seems to be the best solution with I-FGSM.

In summary, some key takeaways are provided as follows. **(i) It is essential to evaluate on a variety of substitute and victim models to gain a comprehensive understanding of the performance of a developed method. (ii) Evaluations using a more advanced optimization back-end (*e.g.*, UN-DP-DI$^2$-TI-PI-FGSM) should be considered. (iii) Generally, using vision transformers as substitute models yields superior attack performance comparing with using the traditional convolutional models. (iv) Among the "substitute model training" methods, the MoreBayesian method consistently enhances adversarial transferability, and it outperforms other methods when using UN-DP-DI$^2$-TI-PI-FGSM as the back-end.**

Except the results on the "validation" models, we further simulate a more practical attack scenario where 15 more victim models that are distinct from those "validation" models in Table 1 are considered. The conclusion remains consistent on attacking these models, except performing LinBP on ConvNeXt achieves the best AA, and we report the results in Section B.

### 4.4    Evaluation of "Generative Modeling" Methods

Another series of transfer-based attacks use generative modeling. They train a generative model to craft adversarial examples and adopt the substitute model as a discriminator. In Table 3, we compare these methods (including CDA [35], GAPF [44], BIA [75], TTP [36], and C-GSP [69]). As the BIA paper also introduce two additional modules (call RN and DA) that could be beneficial under certain circumstances, we also evaluate BIA+RN and BIA+DA and report the results in Table 3. We follow their official implementations and adopted a ResNet-152 model [16] as the substitute model. GAPF performs the best according to our results. For a comprehensive evaluation, we further compare these methods to UN-DP-DI$^2$-TI-PI-FGSM, which is the newly developed optimization back-end. Table 3 shows that it outperforms these generative modeling methods significantly. Moreover, since these methods craft adversarial examples using a generative model, it is infeasible to directly adopt UN-DP-DI$^2$-TI-PI-FGSM for boosting their performance.

Table 3: The performance of generative modeling methods on attacking different victim models. Their substitute model is the same ResNet-152 model. The adversarial examples were generated under an $\ell_\infty$ constraint with $\epsilon = 8/255$. Smaller values indicate more powerful attacks.

| | ResNet-50 | VGG-19 | Inception v3 | EffNetV2-M | ConvNeXt-B | ViT-B | DeiT-B | BEiT-B | Swin-B | Mixer-B | AA |
|---|---|---|---|---|---|---|---|---|---|---|---|
| CDA (2019) [35] | 35.20% | 31.50% | 57.22% | 78.64% | 71.96% | 85.08% | 90.58% | 81.80% | 86.98% | 80.62% | 69.96% |
| GAPF (2021) [44] | 7.22% | **6.20%** | 14.56% | 47.12% | 59.80% | 85.76% | 86.12% | 72.80% | 80.28% | 69.26% | 52.91% |
| TTP (2021) [36] | 44.00% | 32.06% | 59.14% | 89.68% | 91.92% | 94.48% | 95.60% | 92.16% | 94.78% | 85.36% | 77.92% |
| BIA (2022) [75] | 31.26% | 19.16% | 28.40% | 75.24% | 87.12% | 91.24% | 93.90% | 87.12% | 91.78% | 77.20% | 68.24% |
| BIA+RN (2022) [75] | 28.04% | 20.28% | 41.16% | 84.90% | 88.52% | 94.02% | 95.28% | 91.10% | 92.64% | 80.56% | 71.65% |
| BIA+DA (2022) [75] | 44.36% | 28.74% | 43.42% | 82.74% | 88.82% | 90.52% | 93.78% | 85.90% | 92.40% | 76.22% | 72.69% |
| C-GSP (2022) [69] | 52.48% | 43.70% | 70.64% | 91.58% | 91.78% | 90.24% | 95.82% | 84.66% | 94.82% | 86.42% | 80.21% |
| UN-DP-DI$^2$-TI-PI-FGSM | **3.10%** | 7.14% | **10.22%** | **32.38%** | **29.20%** | **49.48%** | **48.46%** | **35.06%** | **50.24%** | **46.14%** | **31.14%** |

## 5 Conclusion

In this paper, we have presented benchmark for transfer-based attacks, called TA-Bench. On TA-Bench, we have implemented 30+ advanced transfer-based attack methods, including those focus on input augmentation and optimizer innovation, those "gradient computation" methods, those "substitute model training" methods, and those applying generative modeling. With TA-Bench, we are capable of evaluating and comparing transfer-based attacks systematically, practically, and fairly. Given comprehensive experimental results on TA-Bench, we have provided new insights about the effectiveness of these attacks, including but not limited to useful combinations of input augmentations and optimizers, reasonable choices of substitute/victim models, *etc.* Hoping to offer a sagacious judge of the state of transfer attacks and help future innovations in this field.

## Acknowledgment

This material is based upon work supported by the National Science Foundation under Grant No. 1801751 and 1956364.

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

# A $\ell_2$ Results

Table 4: Comparing the obtained AA and AAA of some "gradient computation" and "substitute model training" methods. Smaller values indicate more powerful attacks. The adversarial examples were generated under an $\ell_2$ constraint with $\epsilon = 5$.

| | ResNet-50 | VGG-19 | Inception v3 | EffNetV2-M | ConvNeXt-B | ViT-B | DeiT-B | BEiT-B | Swin-B | Mixer-B | AAA |
|---|---|---|---|---|---|---|---|---|---|---|---|
| **I-FGSM Back-end** | | | | | | | | | | | |
| **- Baseline** | | | | | | | | | | | |
| I-FGSM | 88.21% | 92.46% | 94.91% | 97.49% | 89.34% | 91.21% | 90.81% | 90.46% | 95.76% | 95.47% | 92.61% |
| **- Gradient Computation** | | | | | | | | | | | |
| TAP (2018) [77] | 89.06% | 94.55% | 95.42% | 98.39% | 94.51% | 95.26% | 95.68% | 94.90% | 97.19% | 96.77% | 95.17% |
| NRDM (2018) [37] | 91.41% | 92.36% | 96.00% | 98.94% | 95.28% | 97.00% | 97.14% | 97.63% | 97.26% | 95.43% | 95.85% |
| FDA (2019) [11] | 92.64% | 96.62% | 96.10% | 99.16% | 96.95% | 97.72% | 96.66% | 97.00% | 98.15% | 98.54% | 96.95% |
| ILA (2019) [20] | 83.62% | 84.54% | 92.61% | 96.41% | 90.97% | 89.22% | 88.45% | 88.50% | 94.31% | 93.80% | 90.24% |
| SGM (2020) [65] | 79.14% | - | - | **89.40%** | 82.09% | 89.48% | 90.00% | 90.06% | 94.21% | 95.26% | - |
| ILA++ (2020) [26] | 81.01% | 81.77% | 91.44% | 95.83% | 90.42% | 87.86% | 88.90% | 87.40% | 93.78% | 92.41% | 89.08% |
| LinBP (2020) [14] | 84.02% | 90.56% | 97.53% | 98.81% | 91.52% | 92.99% | 92.60% | 93.61% | 96.43% | 98.18% | 93.63% |
| ConBP (2021) [72] | 82.17% | 89.70% | 96.71% | - | - | - | - | - | - | - | - |
| SE (2021) [38] | - | - | - | - | - | 93.67% | 91.12% | 92.79% | - | 95.93% | - |
| FIA (2021) [61] | **74.04%** | **75.87%** | 90.49% | 95.44% | 84.89% | 82.60% | 85.25% | 86.39% | 92.47% | 85.90% | 85.33% |
| PNA (2022) [62] | - | - | - | - | - | 90.56% | 89.32% | 90.18% | 95.17% | - | - |
| NAA (2022) [74] | 79.03% | 85.49% | **88.38%** | 94.84% | **72.61%** | **75.96%** | **77.56%** | **75.04%** | **86.56%** | **84.52%** | **82.00%** |
| **- Substitute Model Training** | | | | | | | | | | | |
| RFA (2021) [47] | 67.24% | - | - | - | - | - | - | - | - | - | - |
| LGV (2022) [13] | 74.86% | - | - | - | - | - | - | - | - | - | - |
| DRA (2022) [78] | **64.29%** | - | - | - | - | - | - | - | - | - | - |
| MoreBayesian (2023) [27] | 70.24% | - | - | - | - | - | - | - | - | - | - |
| **New Optimization Back-end** | | | | | | | | | | | |
| **- Baseline** | | | | | | | | | | | |
| UN-DP-DI$^2$-TI-PI-FGSM | 43.09% | 55.86% | 72.13% | 75.73% | 45.74% | 43.36% | 51.06% | 43.58% | 63.74% | 60.27% | 55.46% |
| **- Gradient Computation** | | | | | | | | | | | |
| TAP (2018) [77] | 77.26% | 65.26% | 82.02% | 91.82% | 52.38% | 70.49% | 78.21% | 53.32% | 83.40% | 72.93% | 72.71% |
| NRDM (2018) [37] | 71.28% | 78.55% | 86.54% | 81.93% | 65.57% | 81.32% | 85.54% | 67.78% | 93.14% | 79.44% | 79.11% |
| FDA (2019) [11] | 58.39% | 65.94% | 78.50% | 96.43% | 79.18% | 98.23% | 95.88% | 83.26% | 95.79% | 96.87% | 84.85% |
| ILA (2019) [20] | 47.80% | 57.54% | 73.57% | 74.58% | 48.88% | 47.89% | 64.93% | 40.11% | 75.82% | 65.62% | 59.67% |
| SGM (2020) [65] | **38.56%** | - | - | **57.71%** | 32.25% | **38.47%** | 36.07% | **33.57%** | **32.94%** | 54.31% | - |
| ILA++ (2020) [26] | 47.58% | 56.46% | 72.96% | 74.80% | 48.95% | 48.26% | 65.76% | 40.88% | 85.21% | 65.55% | 60.64% |
| LinBP (2020) [14] | 48.62% | 56.32% | 89.79% | 97.82% | **30.87%** | 54.97% | 50.57% | 55.38% | 82.27% | 88.77% | 65.54% |
| ConBP (2021) [72] | 46.34% | 56.39% | 83.41% | - | - | - | - | - | - | - | - |
| SE (2021) [38] | - | - | - | - | - | 53.60% | 32.28% | 37.83% | - | **51.99%** | - |
| FIA (2021) [61] | 44.83% | 58.65% | 72.43% | 89.03% | 60.46% | 51.32% | 54.84% | 64.98% | 75.76% | 68.91% | 64.12% |
| PNA (2022) [62] | - | - | - | - | - | 42.10% | **29.33%** | 38.68% | 51.56% | - | - |
| NAA (2022) [74] | 47.22% | 59.99% | 72.75% | 75.64% | 41.47% | 42.20% | 47.26% | 47.91% | 65.69% | 55.74% | 55.59% |
| **- Substitute Model Training** | | | | | | | | | | | |
| RFA (2021) [47] | 57.76% | - | - | - | - | - | - | - | - | - | - |
| LGV (2022) [13] | 41.28% | - | - | - | - | - | - | - | - | - | - |
| DRA (2022) [78] | 64.14% | - | - | - | - | - | - | - | - | - | - |
| MoreBayesian (2023) [27] | **38.90%** | - | - | - | - | - | - | - | - | - | - |

We evaluate the "gradient computation" methods and "substitute model training" methods under $\ell_2$ constraint and provide the results in Table 4. Some $\ell_2$ results are provided in this section. When I-FGSM is applied as the optimization back-end, same as the $\ell_\infty$ results in Table 1 in our main paper, NAA achieves the lowest AAA (*i.e.*, 82.00%) compared with the other "gradient computation" methods, while FIA beats it when ResNet-50 or VGG-19 is chosen as the substitute model. However, unlike in the $\ell_\infty$ setting, SE shows consistently inferior performance when compared with the I-FGSM baseline in the $\ell_2$ setting, and DRA instead of RFA achieves the best performance among "substitute model training" methods.

When UN-DP-DI$^2$-TI-PI-FGSM is applied as the new optimization back-end, same as in the $\ell_\infty$ setting, SGM, PNA, and SE provide favorable attack performance, while PNA on the DeiT-B substitute model turns out to be the best (in the sense of achieving lower BAA) and the generated adversarial examples fools victim models to show an accuracy of only 29.33%. The lowest WAA (which is 51.56%) is obtained by PNA. For the "substitute model training" methods, the MoreBayesian method still outperforms the other methods by a large margin.

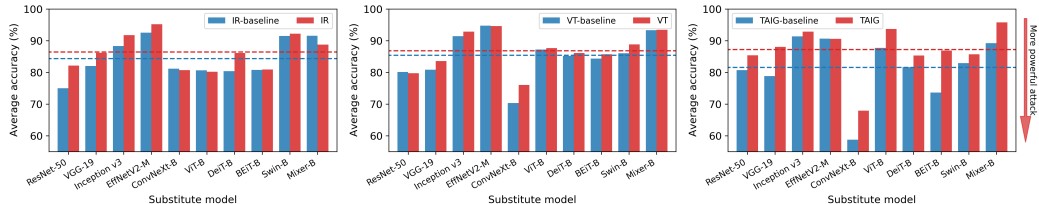

Figure 4: Comparing IR, VT, and TAIG with their corresponding baselines. The dotted lines indicate AAA (lower indicates more powerful attack). The perturbations are constrained under $\ell_2$ norm with $\epsilon = 5$.

Figure 4 compares TAIG, VT, and IR with their corresponding baselines under the $\ell_2$ constraint. It can be seen that the they exhibit higher AAA and worse attack performance compared with their baselines, as in the $\ell_\infty$ setting.

# B    Results of Attacking Other Victim Models

Table 5: Comparing the obtained AA and AAA of some "gradient computation" and "substitute model training" methods on 15 victim models that are distinct from those used to tune the hyper-parameters. Smaller values indicate more powerful attacks. The adversarial examples were generated under an $\ell_\infty$ constraint with $\epsilon = 8/255$.

| | ResNet-50 | VGG-19 | Inception v3 | EffNetV2-M | ConvNeXt-B | ViT-B | DeiT-B | BEiT-B | Swin-B | Mixer-B | AAA |
|---|---|---|---|---|---|---|---|---|---|---|---|
| **I-FGSM Back-end** | | | | | | | | | | | |
| **- Baseline** | | | | | | | | | | | |
| I-FGSM | 89.95% | 91.13% | 95.21% | 96.37% | 89.53% | 93.20% | 93.73% | 92.68% | 95.88% | 95.88% | 93.36% |
| **- Gradient Computation** | | | | | | | | | | | |
| TAP (2018) [77] | 84.03% | 89.09% | 93.36% | 95.72% | 92.93% | 94.66% | 95.38% | 94.66% | 96.54% | 95.73% | 93.21% |
| NRDM (2018) [37] | 83.39% | 85.61% | 88.15% | 97.59% | 96.35% | 96.55% | 96.86% | 96.77% | 96.77% | 93.63% | 93.17% |
| FDA (2019) [11] | 86.43% | 93.09% | 92.23% | 98.69% | 97.36% | 97.51% | 96.94% | 97.69% | 98.09% | 97.93% | 95.60% |
| ILA (2019) [20] | 77.71% | 76.04% | 86.88% | 91.58% | 87.87% | 83.89% | 87.14% | 83.68% | 91.37% | 90.37% | 85.65% |
| SGM (2020) [65] | 76.87% | - | - | 85.64% | 80.16% | 90.84% | 92.02% | 89.59% | 93.42% | 93.90% | - |
| ILA++ (2020) [26] | 75.47% | 73.55% | 91.85% | 89.83% | 86.53% | 81.79% | 86.69% | 82.42% | 90.04% | 88.98% | 84.71% |
| LinBP (2020) [14] | 78.77% | 85.98% | 94.84% | 98.02% | 91.95% | 94.27% | 94.39% | 94.99% | 96.69% | 97.14% | 92.70% |
| ConBP (2021) [72] | 76.61% | 84.77% | 93.30% | - | - | - | - | - | - | - | - |
| SE (2021) [38] | - | - | - | - | - | 93.74% | 93.35% | 93.24% | - | 94.96% | - |
| FIA (2021) [61] | **74.01%** | **72.69%** | 87.48% | 90.45% | 83.36% | 81.52% | 84.98% | 85.15% | 89.70% | 86.07% | 83.54% |
| PNA (2022) [62] | - | - | - | - | - | 92.25% | 92.03% | 91.94% | 95.27% | - | - |
| NAA (2022) [74] | 74.50% | 77.62% | **81.59%** | **86.86%** | **73.31%** | **76.57%** | **78.87%** | **74.60%** | **84.58%** | **84.21%** | **79.27%** |
| **- Substitute Model Training** | | | | | | | | | | | |
| RFA (2021) [47] | **63.93%** | - | - | - | - | - | - | - | - | - | - |
| LGV (2022) [13] | 78.37% | - | - | - | - | - | - | - | - | - | - |
| DRA (2022) [78] | 65.76% | - | - | - | - | - | - | - | - | - | - |
| MoreBayesian (2023) [27] | 68.18% | - | - | - | - | - | - | - | - | - | - |
| **New Optimization Back-end** | | | | | | | | | | | |
| **- Baseline** | | | | | | | | | | | |
| UN-DP-DI$^2$-TI-PI-FGSM | 52.38% | 57.09% | 69.70% | 55.51% | 42.48% | 42.13% | 48.85% | 42.27% | 48.68% | 67.43% | **52.65%** |
| **- Gradient Computation** | | | | | | | | | | | |
| TAP (2018) [77] | 71.84% | 59.21% | 75.30% | 71.15% | 38.75% | 56.97% | 63.83% | 47.73% | 62.06% | 74.42% | 62.13% |
| NRDM (2018) [37] | 60.78% | 63.85% | 77.14% | 63.83% | 51.53% | 64.71% | 74.27% | 61.71% | 82.46% | 76.11% | 67.64% |
| FDA (2019) [11] | 54.96% | 55.67% | 69.87% | 92.98% | 77.22% | 96.78% | 90.49% | 86.03% | 88.45% | 96.69% | 80.91% |
| ILA (2019) [20] | 53.28% | 51.67% | 66.62% | 49.89% | 41.39% | 41.04% | 50.83% | 40.47% | 63.99% | 67.31% | **52.65%** |
| SGM (2020) [65] | **49.87%** | - | - | **39.61%** | 33.54% | **39.80%** | 40.91% | **37.29%** | **31.88%** | 61.79% | - |
| ILA++ (2020) [26] | 53.00% | **51.24%** | **66.27%** | 50.34% | 41.70% | 41.19% | 51.55% | 40.41% | 66.93% | 66.97% | 52.96% |
| LinBP (2020) [14] | 52.97% | 56.00% | 85.62% | 93.87% | **30.81%** | 53.00% | 49.74% | 54.79% | 74.03% | 89.76% | 64.06% |
| ConBP (2021) [72] | 51.69% | 56.04% | 77.79% | - | - | - | - | - | - | - | - |
| SE (2021) [38] | - | - | - | - | - | 49.70% | **36.42%** | 38.79% | - | **61.26%** | - |
| FIA (2021) [61] | 53.63% | 59.96% | 69.64% | 79.53% | 57.92% | 51.26% | 53.32% | 64.99% | 64.75% | 74.17% | 62.92% |
| PNA (2022) [62] | - | - | - | - | - | 44.36% | 37.73% | 41.19% | 36.01% | - | - |
| NAA (2022) [74] | 53.93% | 57.44% | 67.18% | 57.69% | 41.89% | 43.77% | 46.53% | 49.65% | 53.72% | 63.04% | 53.48% |
| **- Substitute Model Training** | | | | | | | | | | | |
| RFA (2021) [47] | 62.08% | - | - | - | - | - | - | - | - | - | - |
| LGV (2022) [13] | 50.79% | - | - | - | - | - | - | - | - | - | - |
| DRA (2022) [78] | 68.80% | - | - | - | - | - | - | - | - | - | - |
| MoreBayesian (2023) [27] | **47.13%** | - | - | - | - | - | - | - | - | - | - |

We collected 15 additional victim models, including 7 CNNs (EfficientNet-L2 [50], ConvNeXt V2-L [64], MobileNet V2 [45], DenseNet-161 [18], ResNeXt-101 [68], SENet-154 [17], and RepVGG-B3 [5]) and 8 vision transformers (ViT-L [8], DeiT-L [54], Swin V2-L [30], BEiT-L [1], CAFormer-B36 [70], MaxViT-L [56], EVA-L [10], EVA02-L [9]), and conducted an experiment on attacking these victim models. For the "substitute model training" methods, the conclusion remains consistent with the observations from Table 1. Specifically, when employing I-FGSM as the back-end, RFA achieves the best AA (*i.e.*, 63.93%), and when applying UN-DP-DI$^2$-TI-PI-FGSM as the back-end, MoreBayesian attains the best AA (*i.e.*, 47.13%). For the "gradient computation" methods, when I-FGSM is applied as the optimization back-end, the conclusion aligns with the findings in Table 1 of the paper. NAA consistently outperforms other methods on most choices of the substitute model, achieving the lowest AAA (*i.e.*, 79.27%). When introducing UN-DP-DI$^2$-TI-PI-FGSM as the optimization back-end, the top four lowest AAs are achieved using ConvNeXt-B, DeiT-B, BEiT-B, and Swin-B as the substitute models, as in Table 1. The best AA is obtained by performing LinBP on the ConvNeXt-B substitute model (*i.e.*, 30.81%, which stands as the second-best in Table 1 and is only 0.20% higher than the best AA), due to slight distribution shift of the tested victim models.

# C   Results of Attacking Robust Models

Table 6: Comparing the obtained AA and AAA of some "gradient computation" and "substitute model training" methods for attacking robust models. The robust victim models include a robust ConvNeXt-B, a robust Swin-B, and a robust ViT-B-CvSt. Smaller values indicate more powerful attacks. The adversarial examples were generated under an $\ell_\infty$ constraint with $\epsilon = 8/255$.

| | ResNet-50 | VGG-19 | Inception v3 | EffNetV2-M | ConvNeXt-B | ViT-B | DeiT-B | BEiT-B | Swin-B | Mixer-B | AAA |
|---|---|---|---|---|---|---|---|---|---|---|---|
| **I-FGSM Back-end** | | | | | | | | | | | |
| *- Baseline* | | | | | | | | | | | |
| I-FGSM | 95.57% | 95.68% | 96.94% | 97.13% | 96.24% | 96.09% | 96.18% | 96.21% | 96.33% | 96.16% | 96.25% |
| *- Gradient Computation* | | | | | | | | | | | |
| TAP (2018) [77] | 95.18% | 95.48% | 96.87% | 97.03% | 96.17% | 96.09% | 96.14% | 96.19% | 96.28% | 96.13% | 96.16% |
| NRDM (2018) [37] | 95.39% | 95.55% | 96.79% | 97.09% | 96.29% | 96.15% | 96.19% | 96.25% | 96.05% | 95.83% | 96.16% |
| FDA (2019) [11] | 94.97% | 95.51% | 96.64% | 97.17% | 96.40% | 96.34% | 96.31% | 96.49% | 96.50% | 96.45% | 96.28% |
| ILA (2019) [20] | 95.39% | 95.55% | 96.68% | 97.04% | 96.08% | 95.75% | 95.87% | 95.91% | 96.00% | 95.79% | 96.01% |
| SGM (2020) [65] | 95.35% | - | - | 96.51% | 95.27% | 95.66% | 95.67% | 95.55% | 95.90% | 95.95% | - |
| ILA++ (2020) [26] | 95.36% | 95.51% | 96.60% | 97.03% | 95.97% | 95.77% | 95.86% | 95.89% | 96.01% | 95.70% | 95.97% |
| LinBP (2020) [14] | 95.33% | 95.56% | 97.00% | 97.10% | 96.21% | 96.10% | 96.15% | 96.17% | 96.33% | 95.99% | 96.19% |
| ConBP (2021) [72] | 95.41% | 95.50% | 97.06% | - | - | - | - | - | - | - | - |
| SE (2021) [38] | - | - | - | - | - | 95.96% | 96.07% | 96.02% | - | 95.97% | - |
| FIA (2021) [61] | 94.81% | **95.21%** | 96.47% | 96.96% | 95.89% | 95.49% | 95.51% | 95.83% | 96.07% | 95.34% | 95.76% |
| PNA (2022) [62] | - | - | - | - | - | 95.86% | 95.94% | 96.05% | 96.27% | - | - |
| NAA (2022) [74] | **94.78%** | 95.31% | **96.01%** | **96.18%** | 93.83% | 94.05% | 93.78% | 94.44% | 95.01% | 94.65% | 94.80% |
| *- Substitute Model Training* | | | | | | | | | | | |
| RFA (2021) [47] | 91.83% | - | - | - | - | - | - | - | - | - | - |
| LGV (2022) [13] | 95.31% | - | - | - | - | - | - | - | - | - | - |
| DRA (2022) [78] | **91.35%** | - | - | - | - | - | - | - | - | - | - |
| MoreBayesian (2023) [27] | 95.21% | - | - | - | - | - | - | - | - | - | - |
| **New Optimization Back-end** | | | | | | | | | | | |
| *- Baseline* | | | | | | | | | | | |
| UN-DP-DI$^2$-TI-PI-FGSM | 94.17% | 95.01% | 96.16% | 96.24% | 94.79% | 93.97% | 93.55% | 93.94% | 94.80% | 94.61% | 94.72% |
| *- Gradient Computation* | | | | | | | | | | | |
| TAP (2018) [77] | 94.77% | 95.34% | 96.52% | 96.57% | 95.22% | 94.95% | 94.73% | 94.79% | 95.43% | 95.31% | 95.36% |
| NRDM (2018) [37] | 95.07% | 95.44% | 96.47% | 96.49% | 95.24% | 95.18% | 95.28% | 95.31% | 95.99% | 95.49% | 95.60% |
| FDA (2019) [11] | 94.70% | 95.07% | 96.21% | 96.96% | 96.19% | 96.62% | 96.16% | 96.05% | 96.02% | 96.79% | 96.08% |
| ILA (2019) [20] | 94.33% | 95.11% | 96.05% | 96.27% | 95.17% | 94.31% | 94.11% | 94.13% | 95.31% | 94.81% | 94.96% |
| SGM (2020) [65] | **93.81%** | - | - | 95.13% | 93.33% | 93.31% | 92.05% | 92.70% | 93.17% | 93.93% | - |
| ILA++ (2020) [26] | 94.37% | 95.01% | 96.01% | 96.22% | 94.97% | 94.19% | 93.79% | 94.07% | 95.60% | 94.63% | 94.89% |
| LinBP (2020) [14] | 94.21% | 95.00% | 96.57% | 96.99% | 94.74% | 94.49% | 93.53% | 94.38% | 95.51% | 95.95% | 95.14% |
| ConBP (2021) [72] | 94.14% | 95.10% | 96.29% | - | - | - | - | - | - | - | - |
| SE (2021) [38] | - | - | - | - | - | 94.04% | 92.94% | 93.81% | - | 94.19% | - |
| FIA (2021) [61] | 94.09% | 94.99% | 95.94% | 96.65% | 95.18% | 94.43% | 93.76% | 94.94% | 95.33% | 94.87% | 95.02% |
| PNA (2022) [62] | - | - | - | - | - | 93.70% | 93.25% | 93.63% | 94.39% | - | - |
| NAA (2022) [74] | 94.15% | **94.97%** | **95.71%** | 95.92% | 94.17% | 93.38% | 92.98% | 93.25% | 94.63% | **93.53%** | **94.27%** |
| *- Substitute Model Training* | | | | | | | | | | | |
| RFA (2021) [47] | **86.37%** | - | - | - | - | - | - | - | - | - | - |
| LGV (2022) [13] | 93.25% | - | - | - | - | - | - | - | - | - | - |
| DRA (2022) [78] | 88.32% | - | - | - | - | - | - | - | - | - | - |
| MoreBayesian (2023) [27] | 92.97% | - | - | - | - | - | - | - | - | - | - |

We also evaluated the performance of different methods in attacking 3 defensive models obtained via adversarial training, *i.e.*, a robust ConvNext-B [29], a robust Swin-B [29], and a robust ViT-B-CvSt [46]. They are all collected from RobustBench [3] and exhibit high robust accuracy against AutoAttack [4]. The results are given in Table 6. It can be seen that when I-FGSM is used as the optimization back-end, NAA and DRA achieve the best AAs among the methods of "gradient computation" and "substitute model training" categories, respectively. When UN-DP-DI$^2$-TI-PI-FGSM is employed as the optimization back-end, SGM achieves the best AAs among the "gradient computation" methods for the most of substitute models, and the best AA is achieved by using DeiT-B as the substitute model, *i.e.*, 92.05%. For the "substitute model training" methods, RFA instead of MoreBayesian achieves the best AA, *i.e.*, 86.37%, since it shares the same training scheme (*i.e.*, adversarial training) with the victim models.

# D   Results of different $\epsilon$

Table 7: Comparing the obtained AA and AAA of some "gradient computation" and "substitute model training". Smaller values indicate more powerful attacks. The adversarial examples were generated under an $\ell_\infty$ constraint with $\epsilon = 16/255$.

| | ResNet-50 | VGG-19 | Inception v3 | EffNetV2-M | ConvNeXt-B | ViT-B | DeiT-B | BEiT-B | Swin-B | Mixer-B | AAA |
|---|---|---|---|---|---|---|---|---|---|---|---|
| **I-FGSM Back-end** | | | | | | | | | | | |
| **- Baseline** | | | | | | | | | | | |
| I-FGSM | 76.99% | 80.93% | 88.83% | 90.96% | 77.34% | 82.21% | 80.46% | 80.56% | 89.18% | 89.61% | 83.71% |
| **- Gradient Computation** | | | | | | | | | | | |
| TAP (2018) [77] | 63.83% | 74.26% | 81.18% | 86.52% | 75.34% | 82.68% | 84.08% | 84.21% | 87.35% | 85.15% | 80.46% |
| NRDM (2018) [37] | 61.20% | 71.48% | 66.88% | 91.78% | 88.04% | 88.84% | 88.28% | 87.02% | 84.87% | 81.27% | 80.97% |
| FDA (2019) [11] | 66.97% | 82.68% | 77.02% | 96.05% | 93.12% | 93.89% | 90.82% | 94.39% | 95.25% | 94.40% | 88.46% |
| ILA (2019) [20] | 53.52% | 54.72% | 66.48% | 81.59% | 71.90% | 56.23% | 62.19% | 58.08% | 75.39% | 63.02% | 64.31% |
| SGM (2020) [65] | 49.48% | - | - | **62.81%** | 49.83% | 72.50% | 74.74% | 67.57% | 78.23% | 81.32% | - |
| ILA++ (2020) [26] | 50.58% | 51.66% | 64.83% | 80.31% | 72.25% | 55.19% | 64.59% | 58.46% | 75.25% | 60.12% | 63.32% |
| LinBP (2020) [14] | 51.00% | 69.69% | 81.37% | 93.56% | 77.26% | 84.32% | 82.03% | 84.74% | 90.36% | 89.58% | 80.39% |
| ConBP (2021) [72] | 46.70% | 66.86% | 74.49% | - | - | - | - | - | - | - | - |
| SE (2021) [38] | - | - | - | - | - | 80.80% | 77.58% | 78.43% | - | 83.42% | - |
| FIA (2021) [61] | 44.43% | 48.30% | 68.02% | 81.63% | 66.31% | 58.58% | 65.40% | 69.84% | 79.10% | 59.90% | 64.15% |
| PNA (2022) [62] | - | - | - | - | - | 76.29% | 71.66% | 76.48% | 85.67% | - | - |
| NAA (2022) [74] | **30.41%** | **44.44%** | **47.31%** | 65.88% | **40.72%** | **34.70%** | 48.45% | 42.29% | 57.48% | **35.02%** | **44.67%** |
| **- Substitute Model Training** | | | | | | | | | | | |
| RFA (2021) [47] | 15.18% | - | - | - | - | - | - | - | - | - | - |
| LGV (2022) [13] | 52.81% | - | - | - | - | - | - | - | - | - | - |
| DRA (2022) [78] | **13.43%** | - | - | - | - | - | - | - | - | - | - |
| MoreBayesian (2023) [27] | 30.89% | - | - | - | - | - | - | - | - | - | - |
| **New Optimization Back-end** | | | | | | | | | | | |
| **- Baseline** | | | | | | | | | | | |
| UN-DP-DI$^2$-TI-PI-FGSM | 15.30% | 25.90% | 34.86% | 34.78% | 18.87% | 13.16% | 20.68% | 14.12% | 27.66% | 21.99% | 22.73% |
| **- Gradient Computation** | | | | | | | | | | | |
| TAP (2018) [77] | 40.72% | 31.08% | 44.02% | 45.43% | 8.07% | 12.47% | 17.29% | 9.02% | 28.59% | 21.79% | 25.85% |
| NRDM (2018) [37] | 22.14% | 32.81% | 43.20% | 41.30% | 20.88% | 24.20% | 33.59% | 16.74% | 61.80% | 18.81% | 31.55% |
| FDA (2019) [11] | 14.84% | 23.93% | 31.10% | 79.50% | 49.80% | 90.39% | 64.71% | 60.18% | 66.88% | 90.22% | 57.16% |
| ILA (2019) [20] | 15.40% | 19.33% | **26.76%** | 25.69% | 13.09% | **8.49%** | 11.18% | 9.19% | 33.32% | 14.46% | **17.69%** |
| SGM (2020) [65] | **12.57%** | - | - | **9.36%** | 6.88% | 10.14% | 8.79% | 7.50% | **7.91%** | 13.67% | - |
| ILA++ (2020) [26] | 14.82% | **18.68%** | 26.84% | 26.64% | 13.81% | 9.11% | 12.27% | 9.53% | 38.84% | 14.91% | 18.54% |
| LinBP (2020) [14] | 15.39% | 24.86% | 61.13% | 76.05% | **5.35%** | 17.55% | 20.55% | 23.06% | 50.74% | 64.17% | 35.89% |
| ConBP (2021) [72] | 14.14% | 25.15% | 46.43% | - | - | - | - | - | - | - | - |
| SE (2021) [38] | - | - | - | - | - | 11.38% | **4.36%** | 5.38% | - | 9.24% | - |
| FIA (2021) [61] | 18.08% | 31.66% | 35.25% | 61.36% | 36.29% | 22.55% | 28.36% | 43.29% | 48.02% | 40.62% | 36.55% |
| PNA (2022) [62] | - | - | - | - | - | 10.34% | 4.60% | 9.26% | 10.30% | - | - |
| NAA (2022) [74] | 15.24% | 23.81% | 27.85% | 35.26% | 16.54% | 12.92% | 18.99% | 25.70% | 34.62% | 18.35% | 22.93% |
| **- Substitute Model Training** | | | | | | | | | | | |
| RFA (2021) [47] | 9.67% | - | - | - | - | - | - | - | - | - | - |
| LGV (2022) [13] | 10.93% | - | - | - | - | - | - | - | - | - | - |
| DRA (2022) [78] | 14.66% | - | - | - | - | - | - | - | - | - | - |
| MoreBayesian (2023) [27] | **6.59%** | - | - | - | - | - | - | - | - | - | - |

We conducted the evaluation of "gradient computation" methods and "substitute model training" methods under an $\ell_\infty$ constraint with $\epsilon = 16/255$ since it is a common setting that many previous work [6, 28, 58, 74] adopted. The results are shown in Table 7. The conclusion is consistent with

observations from Table 1, except when I-FGSM is applied as the optimization back-end, DRA instead of RFA achieves the lowest AA, *i.e.*, 13.43%.

# E  Detailed Results of Input Augmentations and Optimizers

We show the detailed results of different combinations of input augmentations and optimizers in Table 8. It can be seen that UN-DP-DI$^2$-TI-PI-FGSM achieves the best performance on average, despite the optimal solution on different substitute models are different. $\ell_2$ results are given in Table 9.

Table 8: Comparing the obtained AA and AAA of some "gradient computation" and "substitute model training" methods for attacking robust models. The robust victim models include a robust ConvNeXt-B, a robust Swin-B, and a robust ViT-B-CvSt. Smaller values indicate more powerful attacks. The adversarial examples were generated under an $\ell_\infty$ constraint with $\epsilon = 8/255$.

| | ResNet-50 | VGG-19 | Inception v3 | EffNetV2-M | ConvNeXt-B | ViT-B | DeiT-B | BEiT-B | Swin-B | Mixer-B | AAA |
|---|---|---|---|---|---|---|---|---|---|---|---|
| PGD | 88.36% | 91.63% | 93.72% | 95.74% | 88.50% | 90.83% | 90.71% | 89.89% | 94.57% | 94.46% | 91.84% |
| I-FGSM | 87.79% | 91.21% | 93.71% | 95.46% | 88.32% | 90.28% | 90.28% | 89.56% | 94.81% | 94.37% | 91.58% |
| UN-PGD | 86.07% | 88.03% | 93.02% | 94.12% | 83.11% | 89.74% | 89.19% | 88.56% | 92.37% | 94.12% | 89.83% |
| UN-I-FGSM | 85.01% | 86.88% | 93.03% | 94.04% | 82.78% | 89.12% | 89.20% | 87.76% | 91.78% | 93.62% | 89.32% |
| SI-PGD | 86.51% | 86.22% | 91.97% | 89.31% | 83.90% | 88.96% | 85.54% | 87.67% | 92.52% | 92.96% | 88.56% |
| SI-FGSM | 86.21% | 85.79% | 91.74% | 89.63% | 83.87% | 88.79% | 84.78% | 87.18% | 91.87% | 92.79% | 88.26% |
| NI-FGSM | 82.91% | 87.23% | 90.63% | 92.09% | 82.99% | 87.14% | 85.22% | 86.10% | 91.66% | 91.97% | 87.79% |
| PI-FGSM | 82.46% | 87.04% | 90.24% | 91.97% | 82.79% | 87.06% | 85.36% | 85.98% | 91.32% | 92.16% | 87.64% |
| MI-FGSM | 82.42% | 86.94% | 90.44% | 91.91% | 82.99% | 87.14% | 85.27% | 85.86% | 91.36% | 92.04% | 87.64% |
| MI-PGD | 83.20% | 87.59% | 90.97% | 91.47% | 80.93% | 87.07% | 84.40% | 85.62% | 90.87% | 91.71% | 87.38% |
| ...... | | | | | ...... | | | | | | |
| UN-DP-SI-DI$^2$-TI-PI-PGD | 42.88% | 50.34% | 60.68% | 44.19% | 32.34% | 37.28% | 39.33% | 35.56% | 46.66% | 44.47% | 43.37% |
| UN-DP-SI-DI$^2$-TI-NI-FGSM | 42.78% | 50.40% | 60.59% | 44.10% | **32.33%** | 36.93% | 39.42% | 35.83% | 46.37% | **44.22%** | 43.30% |
| UN-DP-SI-DI$^2$-TI-MI-FGSM | 42.85% | 50.34% | 60.42% | 44.03% | 32.49% | 36.73% | 39.30% | 35.91% | 46.52% | 44.31% | 43.29% |
| UN-DP-SI-DI$^2$-TI-PI-FGSM | 42.92% | 50.12% | 60.55% | **44.00%** | 32.47% | 36.74% | 39.57% | 35.94% | 46.16% | 44.30% | 43.28% |
| UN-DP-DI$^2$-TI-PI-PGD | 35.68% | 49.07% | 59.48% | 52.40% | 33.56% | 33.53% | **35.58%** | 34.85% | 45.92% | 46.30% | 42.64% |
| UN-DP-DI$^2$-TI-MI-PGD | 35.57% | 48.70% | 59.34% | 52.34% | 33.66% | 33.69% | 35.75% | 34.84% | 45.78% | 46.45% | 42.61% |
| UN-DP-DI$^2$-TI-NI-PGD | **35.34%** | 48.55% | 59.19% | 52.20% | 33.39% | 33.39% | 35.72% | 34.83% | 45.71% | 46.42% | 42.47% |
| UN-DP-DI$^2$-TI-MI-FGSM | 35.80% | 48.86% | 59.15% | 52.67% | 33.22% | 33.19% | 35.90% | 34.14% | 45.28% | 46.34% | 42.46% |
| UN-DP-DI$^2$-TI-NI-FGSM | 35.74% | 48.77% | 59.06% | 52.70% | 33.16% | 33.26% | 35.68% | 34.24% | 45.46% | 46.40% | 42.45% |
| UN-DP-DI$^2$-TI-PI-FGSM | 35.70% | **48.33%** | **58.62%** | 52.98% | 33.64% | **32.74%** | 36.58% | **33.72%** | **45.24%** | 46.60% | **42.42%** |

Table 9: Detailed results of different combinations of input augmentations and optimizers. Smaller values indicate more powerful attacks. The adversarial examples were generated under an $\ell_2$ constraint with $\epsilon = 5$.

| | ResNet-50 | VGG-19 | Inception v3 | EffNetV2-M | ConvNeXt-B | ViT-B | DeiT-B | BEiT-B | Swin-B | Mixer-B | AAA |
|---|---|---|---|---|---|---|---|---|---|---|---|
| PGD | 88.24% | 92.73% | 94.87% | 97.34% | 89.59% | 90.66% | 91.39% | 90.26% | 95.74% | 95.16% | 92.60% |
| NI-FGSM | 87.44% | 91.47% | 94.51% | 96.70% | 90.03% | 91.34% | 90.56% | 91.02% | 95.40% | 95.26% | 92.37% |
| NI-PGD | 87.37% | 91.80% | 94.84% | 96.74% | 89.22% | 91.27% | 90.74% | 91.06% | 95.29% | 95.38% | 92.37% |
| MI-PGD | 87.64% | 91.64% | 94.60% | 96.59% | 89.27% | 91.31% | 90.86% | 91.02% | 95.07% | 95.48% | 92.35% |
| MI-FGSM | 87.18% | 91.39% | 94.49% | 96.63% | 89.89% | 91.53% | 90.60% | 91.10% | 95.21% | 95.33% | 92.34% |
| I-FGSM | 87.93% | 91.82% | 94.76% | 97.24% | 88.96% | 91.01% | 90.64% | 90.18% | 95.46% | 95.10% | 92.31% |
| PI-PGD | 87.52% | 91.63% | 94.67% | 96.69% | 89.03% | 91.04% | 90.73% | 91.08% | 95.14% | 95.36% | 92.29% |
| PI-FGSM | 87.20% | 91.14% | 94.37% | 96.69% | 89.92% | 91.36% | 90.50% | 90.79% | 95.34% | 95.27% | 92.26% |
| UN-PGD | 86.56% | 89.87% | 94.66% | 96.82% | 86.18% | 90.82% | 90.68% | 89.94% | 94.42% | 95.18% | 91.51% |
| UN-I-FGSM | 86.56% | 89.64% | 94.38% | 96.73% | 85.83% | 90.39% | 90.19% | 89.26% | 94.01% | 94.86% | 91.18% |
| ...... | | | | | ...... | | | | | | |
| UN-DP-SI-DI$^2$-TI-NI-PGD | 52.20% | 59.40% | 74.95% | **70.55%** | 42.80% | 50.19% | 52.94% | 44.73% | 63.32% | 55.88% | 56.70% |
| UN-DP-SI-DI$^2$-TI-NI-FGSM | 52.17% | 59.24% | 75.14% | 70.60% | **42.63%** | 50.24% | 52.55% | 44.73% | 63.63% | **55.68%** | 56.66% |
| UN-DP-DI$^2$-TI-PGD | 46.81% | 58.84% | 74.67% | 74.50% | 43.99% | 46.13% | 50.09% | **43.75%** | 61.23% | 63.36% | 56.34% |
| UN-DP-DI$^2$-TI-FGSM | 46.50% | 59.08% | 74.34% | 74.19% | 43.64% | 45.81% | **49.83%** | 43.80% | **60.22%** | 63.54% | 56.09% |
| UN-DP-DI$^2$-TI-MI-PGD | 42.93% | 55.49% | 72.58% | 74.89% | 45.58% | 45.01% | 51.43% | 44.97% | 64.73% | 61.06% | 55.87% |
| UN-DP-DI$^2$-TI-MI-FGSM | 42.72% | 55.89% | 72.71% | 74.67% | 45.28% | 44.77% | 51.01% | 44.84% | 64.49% | 61.20% | 55.76% |
| UN-DP-DI$^2$-TI-NI-PGD | 42.70% | 55.82% | 72.46% | 74.84% | 45.23% | 44.81% | 51.10% | 44.43% | 64.53% | 61.08% | 55.70% |
| UN-DP-DI$^2$-TI-PI-PGD | 42.91% | 55.32% | 72.65% | 74.76% | 45.38% | 44.90% | 51.43% | 44.64% | 64.45% | 60.58% | 55.70% |
| UN-DP-DI$^2$-TI-NI-FGSM | **42.44%** | 56.09% | 72.53% | 74.70% | 45.14% | **44.22%** | 50.63% | 44.59% | 64.46% | 61.36% | 55.62% |
| UN-DP-DI$^2$-TI-PI-FGSM | 43.01% | **55.46%** | **72.46%** | 74.63% | 45.17% | 44.74% | 51.34% | 44.53% | 64.21% | 60.51% | **55.61%** |

# F   Implementation Details

**Augmentations and Optimizer.** For PGD, DI$^2$-FGSM, MI-FGSM, NI-FGSM, and PI-FGSM, we use the default hyper-parameters. For TI-FGSM, we randomly translate the input with a range of [-3, +3] since its performance is better than the approximation using a $7 \times 7$ Gaussian kernel in many implementations [28, 58, 59, 27]. For SI-FGSM and Admix, both of them average the gradients obtained by feeding different augmented inputs into the substitute model, which may lead to unfair comparisons. Therefore, we randomly select one input from the augmented copies, and the hyper-parameters remain the same as in their original papers. For UN, the noise added to the input follows $\mathcal{U}(-\epsilon, \epsilon)$ and $\mathcal{U}(-\frac{\epsilon}{\sqrt{HW}}, \frac{\epsilon}{\sqrt{HW}})$ (the dimension of inputs is $3 \times H \times W$) for attacks under $\ell_\infty$ and $\ell_2$ constraints, respectively. For DP, we divide the perturbation into $16 \times 16$ patches and randomly drop 50% of the patches at each iteration.

**Gradient Computation.** For TAIG, VT, IR, TAP, FDA, SE, and PNA, we set the same hyper-parameters as in their original papers. For NRDM, ILA, ILA++, LinBP, ConBP, FIA, and NAA, the main hyper-parameter which significantly impacts the performance is the choice of the middle layer. The scaling factor of SGM is also related to the selection of the substitute model. We tune these hyper-parameters by evaluating on a validation set consisting of 500 samples that do not overlap with the samples in the test set, and the best choices are shown in Table 10.

Table 10: The index of the middle layer we chose for each substitute model for each method. For VGG-19, max-pooling layers were considered as the end of blocks.

| | ResNet-50 | VGG-19 | Inception v3 | EffNetV2-M | ConvNeXt-B | ViT-B | DeiT-B | BEiT-B | Swin-B | Mixer-B |
|---|---|---|---|---|---|---|---|---|---|---|
| **I-FGSM Back-end** | | | | | | | | | | |
| NRDM | 2 | 5 | 5 | 3 | 4 | 8 | 6 | 2 | 2 | 4 |
| ILA | 2 | 5 | 5 | 3 | 1 | 4 | 6 | 4 | 2 | 2 |
| ILA++ | 2 | 5 | 5 | 3 | 1 | 4 | 6 | 4 | 2 | 2 |
| LinBP | 3 | 8 | 3 | 6 | 4 | 11 | 11 | 11 | 4 | 4 |
| ConBP | 3 | 8 | 3 | - | - | - | - | - | - | - |
| FIA | 2 | 5 | 5 | 3 | 1 | 2 | 4 | 4 | 2 | 2 |
| NAA | 2 | 5 | 5 | 3 | 1 | 4 | 4 | 4 | 2 | 2 |
| **New Optimization Back-end** | | | | | | | | | | |
| NRDM | 3 | 6 | 6 | 7 | 4 | 12 | 12 | 10 | 4 | 10 |
| ILA | 4 | 7 | 6 | 7 | 4 | 12 | 10 | 12 | 2 | 10 |
| ILA++ | 4 | 7 | 6 | 7 | 4 | 12 | 10 | 12 | 4 | 10 |
| LinBP | 4 | 9 | 7 | 6 | 4 | 11 | 11 | 11 | 4 | 11 |
| ConBP | 4 | 9 | 7 | - | - | - | - | - | - | - |
| FIA | 4 | 8 | 7 | 6 | 4 | 12 | 12 | 12 | 4 | 12 |
| NAA | 4 | 8 | 6 | 6 | 3 | 10 | 12 | 10 | 4 | 8 |

**Substitute Model Training.** In this category of methods, ResNet-50 is commonly chosen as the substitute model, and we collect the models from the GitHub repositories of these methods. For LGV and the MoreBayesian method, we only sample once at each iteration.

**Generative Modeling.** In this category of methods, all the generators are collected from the GitHub repositories of these methods.

