# OpenReview forum: "Towards Evaluating Transfer-based Attacks Systematically, Practically, and Fairly"
_NeurIPS.cc/2023/Conference — NeurIPS 2023 poster_

### Official Review · Reviewer_NrCP · 2023-06-29

**Soundness:** 3 good
**Presentation:** 3 good
**Contribution:** 4 excellent
**Rating:** 7
**Confidence:** 2

**Summary:**

This paper introduces a benchmark, called TA-Bench, for transfer-based attacks. The authors implement 30+ transfer-based attack methods that are mostly proposed in the last 3 years. This paper takes several aspects of transfer-based attacks into consideration, including augmentation, optimizer, substitute model training, and generative modeling.

**Strengths:**

1. Solid work with a considerable amount of experiments. I believe that this work, as its topic suggests, will bring new insights in systematically, practically, and fairly evaluating transfer-based attacks.
2. Apart from the benchmark, this paper provides some useful takeaways in Line 338-345.

**Weaknesses:**

This paper would better be submitted to the benchmark track.

**Questions:**

1. The recent paper "Reliable Evaluation of Adversarial Transferability" by Yu et al. also provides a benchmark for evaluating the transferability of adversarial examples. Could you please compare your work with theirs?

**Limitations:**

No.

---

> ### Author Rebuttal · Authors · 2023-08-09
>
> Thanks for the positive feedback. Except for the comment about "submitting to the benchmark track" which is addressed in our general response, all the comments are replied to as follows.
> &nbsp;
>
> > The recent paper "Reliable Evaluation of Adversarial Transferability" by Yu et al. also provides a benchmark for evaluating the transferability of adversarial examples. Could you please compare your work with theirs?
>
> **A:** We appreciate the point to this paper. It seems that this paper is available online after the NeruIPS submission deadline, thus it was not discussed in the paper. Comparing with it, the advantages of our TA-Bench include at least:
> * More evaluated methods.
> * More pairs of substitute-victim models (forming more comprehensive evaluations).
> * A new and advanced back-end optimization method, _i.e._, UN-DP-DI$^2$-TI-PI-FGSM.

---

> > ### Comment · Reviewer_NrCP · 2023-08-16
> >
> > Thank the authors for their response. The rebuttal and global response have fully addressed our concerns and we have no follow-up questions. We will keep our score and recommend accepting this paper.

---

> > > ### Author Response · Authors · 2023-08-16
> > > **Thanks to the reviewer**
> > >
> > > Dear Reviewer NrCP,
> > >
> > > We are pleased to know that your concerns have been fully addressed and thank you for recommending the acceptance of our paper!
> > >
> > > Best regards,
> > > Authors

---

### Official Review · Reviewer_QeCx · 2023-07-05

**Soundness:** 3 good
**Presentation:** 3 good
**Contribution:** 4 excellent
**Rating:** 6
**Confidence:** 2

**Summary:**

This paper establishes a transfer-based attack benchmark (TA-Bench) so that researchers could take advantage of this to comare different methods systematically, fairly and practically. TA-Bench implements 30+ methods and evaluate on 10 popular victim models (architecture) on ImageNet.

**Strengths:**


1.The paper is well-written and easy to follow.

2.The TA-Bench has practical value for transfer-based attacks, since it implements 30+ methods and provides comprehensive evaluations.


**Weaknesses:**



1.The benchmark focuses on ImageNet. Is it possible to extend to other datasets, smaller (MNIST) or larger (JFT300)?

2.Typos:
1)The term “TA-bench” and “TA-Bench” should be consistent.



**Questions:**


1.This paper did a very good job. TA-Bench is comprehensive. I score 6 for this paper. But this paper may fit better towards “NeurIPS Tack Datasets and Benchmarks”.
2.In table 1, what is the meaning of different colors, e.g., grey, black, bald black?


**Limitations:**


The authors adequately addressed the limitations.

---

> ### Author Rebuttal · Authors · 2023-08-09
>
> Thanks for the positive feedback. Except for the the questions about the the D&B track which is answered in our general response, all your comments are replied as follows.
> &nbsp;
>
> > The benchmark focuses on ImageNet. Is it possible to extend to other datasets, smaller (MNIST) or larger (JFT300)?
>
> **A:** We focus on ImageNet first for several reasons. First of all, almost all papers studying transfer-based attacks **developed** and **evaluated** their methods on ImageNet. Only a few papers conducted evaluations on smaller scale datasets, such as CIFAR-10, as well. This is because ImageNet contains large-scale diverse images, making the data distribution more representative. We ensure a fair comparison on ImageNet first to benefit understanding what's effective. Another reason why we focus on ImageNet first is that vision transformers generally require training on ImageNet, and the dataset offers more options for choosing substitute/victim models, enabling a more comprehensive understanding of the performance of an attack, as discussed in Section 4.3. After consolidating all results on ImageNet, we will consider performing evaluations on CIFAR-10, too. MNIST is too toy to be used for adversarial experiments now. As for JFT-300M, to the best of our knowledge, it is not publicly available yet.
> &nbsp;
>
> > Typos: 1)The term “TA-bench” and “TA-Bench” should be consistent.
>
> **A:** Thanks for pointing out these typos. We will fix them in an updated version of the paper.
> &nbsp;
>
> > In table 1, what is the meaning of different colors, e.g., grey, black, bald black?
>
> **A:** In Table 1, the number is colored to be grey once the performance is worse than the back-end attack (_i.e._, I-FGSM and UN-DP-DI$^2$-TI-PI-FGSM). The bold black indicates the best performance among all methods in the same category. We will highlight this information in the paper.
> &nbsp;

---

> > ### Comment · Reviewer_QeCx · 2023-08-14
> >
> > Thanks for your responses. I am satisfied with the results.

---

> > > ### Author Response · Authors · 2023-08-15
> > >
> > > Dear Reviewer QeCx,
> > >
> > > We would like to thank you for the positive feedback and for responding to our rebuttal!
> > >
> > > Best regards,
> > > Authors

---

### Official Review · Reviewer_Yj8S · 2023-07-06

**Soundness:** 2 fair
**Presentation:** 2 fair
**Contribution:** 3 good
**Rating:** 4
**Confidence:** 5

**Summary:**

This paper explores the problem of adversarial transferability evaluation on image classification tasks. The authors feel that there are a large number of migration attacks, but this community lacks a standard benchmark. Therefore, this paper establishes a transfer-based attack benchmark (TA-Bench) which implements 30+ methods and then evaluate and compare them comprehensively on 10 popular substitute/victim models on ImageNet.

**Strengths:**

1. The benchmark of this paper integrates many transfer attacks.

2. TA-bench helps researchers to carry out research more easily and in-depth.

**Weaknesses:**

1. We acknowledge the workload and potential of this paper, but compared to previous work, the proposed TA-bench does not consider adversarial defense methods at all, which leads to the wrong robustness assessment. We think that some of the latest pre-processing defense and adversarial training models should be taken into account, at least ens3-adv-Inception-v3, ens4-adv-Inception-v3, ens-adv-Inception-ResNet-v2, HGD, R&P, NIPS-r3, JPEG, FD, ComDefend, NRP, RS, Bit-Red, DiffPure). We will not give references here, because it is very common in this community.

2. The scalability of TA-bench is not strong enough. Transfer attacks have recently paid more attention to the research of targeted attacks and are a future research direction, but TA-bench lacks corresponding evaluations, which restricts its contribution. We believe that the category relationship of images should be considered in the screening of data sets, and an evaluation system for targeted transfer attacks should be constructed.

3. The motivation for this paper is not very sufficient. The set of models in the NIPS2017 adversarial competition is indeed relatively limited, but I-FGSM does show the most powerful attack performance as an optimized backend (after integrating with other methods), and the experiments in this paper also illustrate this point. We think it is best for the authors to start with the challenge of transfer attacks to build benchmarks, including but not limited to targeted attacks, ViT to CNN.

4. More and more latest foundation models adopt the transformer architecture, but in this paper, the discussion of the adversarial transferability of ViT and CNN is not very sufficient. This is also a recent research hotspot.

5. Also, we are a little bit less sympathetic to the taxonomy of this paper. We feel that it can be divided into more detailed classifications such as advanced optimization, data enhancement, model integration, feature attack, and network structure, rather than just "Gradient Computation" Methods and "Substitute Model Training" Methods".

**Questions:**

See Weaknesses.

**Limitations:**

The authors did not discuss limitation and ethical risks in the main body.

---

> ### Author Rebuttal · Authors · 2023-08-09
>
> Thanks for the feedback. Except for the questions about defense methods, targeted attacks, and the taxonomy which are answered in our general response, all your comments are replied to as follows.
> &nbsp;
>
> > We think it is best for the authors to start with the challenge of transfer attacks to build benchmarks.
>
> **A:** TA-Bench is motivated by the fact many existing transfer-based papers show a lack of systematically, fairly, and practically experimental evaluations. First of all, different architectures should be considered when setting up the substitute and the victim models, while in many recent papers, only convolutional networks were considered. Second, many methods verified their effectiveness only with a very basic optimization back-end, _i.e._, I-FGSM, despite the existence of more advanced optimizers and input augmentations. It is hence unclear whether the technical innovation really works in practice when consolidating all efforts in transfer-based attacks. We believe these problems hampered the development of adversarial machine learning and it motivates us to establish such a benchmark (_i.e._, TA-Bench) for evaluating transfer-based attacks.
>
> We agree that although bootstrapping a challenge is a point to establish a benchmark, our motivation also well shows the necessity of building the benchmark.
> &nbsp;
>
> > The discussion of the adversarial transferability of ViT and CNN is not very sufficient.
>
> **A:** In our discussions about cross-architecture transferability (in Section 2 in our supplementary material), we have shown the performance when adopting transformers/CNNs as the substitute model, as the victim model, or both. Besides the tested transformer models and convolutional models, it is easy to incorporate more models in our benchmark.

---

> > ### Comment · Reviewer_Yj8S · 2023-08-17
> >
> > In previous work, we always test the performance of defense methods, which is crucial for evaluating adversarial robustness. Likewise, targeted attacks pose a more significant threat to real-world applications. We still believe that this benchmark will definitely lead to the development of the field, but these two items are still not to be ignored. These are all available in the data set of our commonly used NIPS2017 adversarial competition. So we still need this work to enrich these contents further. Our scores will not affect the acceptance of this paper, but we still hope that the authors can implement these to guide the correct adversarial robustness evaluation.

---

> > > ### Author Response · Authors · 2023-08-17
> > >
> > > Dear Reviewer Yj8S,
> > >
> > > Thanks for responding to us and recognizing our contributions as "definitely lead to the development of the field".
> > > For the comment about defense, we would like to gently remind that, as reported in our general response, the experiment on defensive models has been carried out and many results have already been given in the attached pdf in the general response. We will also incorporate these results into our paper as mentioned. As for targeted attacks, we will test them when appropriate, as suggested.
> > >
> > > Best regards,
> > > Authors

---

> ### Author Response · Authors · 2023-08-17
> **Your further feedback would be greatly appreciated.**
>
> Dear Reviewer Yj8S,
>
> Thanks again for your comments! We have detailedly provided responses to all your comments. If there are any remaining concerns about our paper, we are more than delighted to address them. Have a nice day.
>
> Best regards,
> Authors

---

### Official Review · Reviewer_fXNW · 2023-07-08

**Soundness:** 3 good
**Presentation:** 2 fair
**Contribution:** 3 good
**Rating:** 5
**Confidence:** 5

**Summary:**

The paper proposes a new benchmark of techniques designed to increase the transferability of adversarial examples. The paper implements more than 30 of these techniques to compare the success rate of the corresponding attacks. The paper identifies several flaws of current evaluation protocols, for example, not considering the pre-processing pipeline.

**Strengths:**

- The engineering work done is impressive. Benchmarking 30 techniques for transferability is a testimony of good view of the field of transferability, and of a strong software development work.
- The paper correctly and fairly evaluates techniques, regarding best practices developed recently in \[59\]. In particular, the paper takes a particular care to control the effect of the number of gradients computed per iteration, which should be controlled for fairness.
- Some choices in categorisation are sound. For example, evaluating separately techniques based on generative modelling makes sense.
- The related work is sensible and extensive. For example, the paper correctly acknowledges and iterate about the most related work \[59\], another benchmark paper on transferability. Unfortunately, the review of techniques is not exhaustive. And the paper would benefit from a clear explanation on how the 30 methods were selected. For example, the paper could exhaustively list all the techniques for transferability published at top-conferences.
- The paper proposes a more consistent way to evaluate the complementary of techniques than what is used currently. But there are some limitations (see below).

**Weaknesses:**

- An exhaustive review of techniques published at top-conferences (as done for review papers) would clarify why some techniques are included and not other. For example, the paper titled "Understanding and Enhancing the Transferability of Adversarial Examples" was the first to average gradients over (Gaussian) noise in 2018. Moreover, "Learning Transferable Adversarial Examples via Ghost Networks" (AAAI-20), "On Success and Simplicity: A Second Look at Transferable Targeted Attacks" (NeurIPS 2021), "Efficient and Transferable Adversarial Examples from Bayesian Neural Networks" (UAI 2022), "Boosting the Transferability of Adversarial Attacks with Reverse Adversarial Perturbation" (NeurIPS 2022) could be considered.
- No code is provided as supplementary materials. It is impossible to evaluate the claimed contribution of a unified codebase. Open-sourcing the code is particularly important for benchmark papers, where its value depends on the cleanness and extensibility of its code base.
- The usage of cross-validation is weak and not systematic. I think that the benchmark should provide a standardised protocol to tune hyperparameters and select any elements with cross-validation. For example, the paper should select the best combination of augmentation and optimizer by cross-validation (section 4.2). This is a significant flaw of the current paper. Some tuning of the hyperparameters is currently done by on a set of hold-out target models (Section 4.3). But these models are removed from the test set of targets models. Therefore, there is an overlap between validation and targets between the experiments. The test evaluation is weaker because performed on fewer models. It would be better to use additional models that would be used only for validation. As the code is not shared, I cannot check if the codebase includes an easy way to tune the hyperparameters of any techniques implemented in the benchmark. From my experience, tuning the hyperparameters is of first importance when changing some experimental settings. In particular, selecting the SGM hyperparameter when changing the source architecture.
- A limitation of the current benchmark is that the training dataset is supposed to be known. I think this overestimates significantly the success rates (I suspect that the effect is larger than the one of the pre-processing). I understand removing this hypothesis is very costly computationally. But the paper should therefore not position itself as a more realistic evaluation. In addition, I think training and distributing source and target models without this hypothesis would add a high value to the benchmark (in practice, splitting ImageNet in two or introducing some distributional shift between the datasets).
- The paper does not extend the evaluation of some techniques to other source architecture. For example, in Table 1, most columns are empty for substitute model training. I understand that some techniques are too costly to train from scratch (RFA for example), but LGV and MoreBayesian should be feasible since they require only a few additional epochs from a pretrained model.
- The categories of techniques used are debatable. For example, I think that LGV and MoreBayesian should be categorised as gradient computation. Both techniques are training-based, but their primary objective is *not* to train a better single base substitute model, but to augment an existing base pretrained model and obtain several slightly modified models (to be used one per gradient). The "augmentation" category should be renamed "input augmentation" or "data augmentation" to clarify. I am not sure why input augmentation techniques and optimizers techniques are evaluated together as a single category of techniques, and "gradient computation" techniques are not evaluated together with optimizers too. In fact, numerous "gradient computation" techniques also augment the surrogate model (LinBP, GN, SGM, etc.). I think that a more granular evaluation of the category of techniques, category per category, as done in \[59\], would be more advisable.
- I think that this paper would have been a perfect fit for the benchmark and datasets track of NeurIPS. I feel that the impact of the paper is slightly too limited for the main track. The conclusions of the benchmark have a somehow limited significance for the community.
- The paper evaluates the gradient computation category on the best combination of augmentation and optimizer techniques. It would be good to evaluate the relation in both ways: the gradient computation category on I-FSGM vs. the gradient computation category on the best surrogate models.

  **Minor comments**

- The paper would benefit from polished writing and improved formatting. Bold should be used with care, only for sparse important keywords. It would be better to highlight entire sentences (or paragraph) with italic instead of bold. The writing should be improved overall. For example,  "Similar for IR and TAIG." (l.272) is not a valid sentence, "state-of-the-arts" (l.183) is not a valid word. Exaggeration must be avoided. For example, please avoid familiar and exaggerated formulations like "super close" (l.329). The paper would benefit from a more precise writing. For example, the paper states that "30+" techniques are compared. The exact number of techniques is never mentioned. Some legends are missing some descriptions. For example, the legend of Table 1 does not specify what the grey colour means, and does not specify that the first row is the source model.
- No evaluation of defended target models is performed. This is not necessarily a major issue, but it is advisable to discuss this limitation. Same for targeted/untargeted perturbations.
- I think that the paper would be more clear if it reports success rates instead of accuracies.
- Specifying the step size of I-FGSM relatively to the epsilon norm of the perturbation would be better and improve consistency. For example, use alpha = eps/10 for both Linf and L2 norms.

**Questions:**

- Could you share the code base (anonymously) to review it?
- Which hyperparameters are currently selected by cross-validation? More details should be provided on how did you tune the hyperparameters (paragraph l.228-l.241).
- Will you publish the code to sample the 5K subset of examples? This code is needed since this subset would need to be rebuilt for each new model training techniques, to ensure fairness. The 5K examples are correctly classified by all existing surrogate models, so it should be the case for    surrogate models that will be added later. It is important to specify this in the documentation.
- Which number of random restarts did you use for PGD? I did not find this information in the paper.
- Line 324 states that transformers are better surrogate architecture than CNN. Is it true for all types of targets? I.e. does this observation hold for the BAA (best-case accuracy)? Or is it simply related to the fact that there are more transformers in the set of target models (and we are confused by the average AAA)?
- Can you describe more precisely the training differences, briefly mentioned line 332?

I am ready to increase my score depending on how the authors answer the weaknesses and questions.

**Limitations:**

Some limitations are not discussed (see above weakness section)

---

> ### Author Rebuttal · Authors · 2023-08-09
>
> Thanks for the positive feedback. Except for common questions which are answered in our general response, all your comments are relied to as follows.
> &nbsp;
>
> > Unfortunately, the review of techniques is not exhaustive.
>
> **A:** Thanks for pointing out these methods. The implementation of "UN" in our paper is equivalent to one of the mentioned methods, and we will highlight it in the revised paper. Evaluation results of the other mentioned methods and several very recent methods (that are available online after the NeurIPS submission deadline) will be added to the paper.
> &nbsp;
>
> > Hyper-parameters and a standard protocol to tune them.
>
> **A:** Our benchmark does have a protocol to tune hyper-parameters. All important hyper-parameters, including the choice of position for NRDM, ILA, ILA++, LinBP, ConBP, FIA, and NAA, and the scaling factor for SGM (as you have mentioned) can all be tuned on a validation set that consists of 500 examples that do not overlap with the test data. The information will be highlighted in an updated version of our paper. The validation set and the tuning protocol will also be made publicly available.
> &nbsp;
>
> > A limitation of the current benchmark is that the training dataset is supposed to be known.
>
> **A:** As mentioned in the paper, we consider such a threat model just to keep in line with previous work that developed these transfer-based attacks. To the best of our knowledge, all the compared methods, in their original papers, took a substitute model trained on the same dataset as that used to train the victim models for experiments. While we fully agree that evaluating under a more stringent threat model is insightful, considering that we have made obvious changes to the experiments (including introducing various types of substitute/victim models, adopting more realistic pre-processing, _etc_), further modifying the models into ones trained on independent datasets **might lead to confusion about what leads to a performance change of these compared methods**. In addition, performing such an experiment requires determining many critical factors, such as the number of training images that the attacker is able to collect and even the resources the attacker owns. There are some papers that were written to deal with these problems, and we are more than glad to consider such an experiment in future work if possible.
> &nbsp;
>
> > In Table 1, most columns are empty for substitute model training.
>
> **A:** The reason why we did not extend the evaluation of "substitute model training" methods is indeed about training cost. Especially, training for these methods requires tuning many additional hyper-parameters, such as learning rate, weight decay, batch size, and $\lambda$ and $\gamma$ in MoreBayesian. Thus, the training cost is high even for LGV and MoreBayesian.
> &nbsp;
>
> > The gradient computation category on I-FSGM and the gradient computation category on the best surrogate models.
>
> **A:** We have reported the results of the "gradient computation" methods on I-FGSM in Table 1 (upper half), and these results can be easily compared with those on UN-DP-DI$^2$-TI-PI-FGSM. Table 1 also provides results on each surrogate model, thus the performance on the best surrogate models is given as expected.
> &nbsp;
>
> > The paper would benefit from polished writing and improved formatting.
>
> **A:** Thanks for the kind suggestions. We shall improve the writing and formatting accordingly.
> &nbsp;
>
> > I think that the paper would be more clear if it reports success rates instead of accuracies.
>
> **A:** As we discussed in lines 168-171 in our paper, we adopt prediction accuracy in evaluating attack performance because it is easier to incorporate other substitute/victim models in the future, since a reasonable calculation of the attack success rate requires benign examples to be correctly classified by all victim models, as suggested by previous papers.
> &nbsp;
>
> > The step size of I-FGSM.
>
> **A:** We follow the setting of using a step size of 1/255 for $\ell_\infty$ attacks, as in many previous papers.
> &nbsp;
>
> > Will you publish the code to sample the 5K subset of examples?
>
> **A:** Certainly, we will publish the code and we will further provide the same 5K examples for future evaluations to the public.
> &nbsp;
>
> >  Which number of random restarts did you use for PGD?
>
> **A:** We didn't perform random restarts for PGD. Performing random restarts by $k\times$ times (with a fixed number of a maximal number of iterations) requires reducing the number of iterations within each run by $k\times$, and we observe no performance gain in such a setting.
> &nbsp;
>
> > Line 324 states that transformers are better surrogate architecture than CNN. Is it true for all types of targets?
>
> **A:** From Table 4 in our supplementary material, we can see that ViT-B shows better cross-architecture adversarial transferability in the sense of worst-case attack performance and average performance (over 5 CNNs, 4 transformers, and an MLP). Yet, in the sense of the best-case performance, ResNet-50 seems better.
> &nbsp;
>
> > Can you describe more precisely the training differences, briefly mentioned line 332?
>
> **A:** ViT-B used the private JFT dataset for pre-training, and it was trained on ImageNet after pre-training. DeiT-B was trained on ImageNet using a variety of heavy data augmentation and regularizations. BEiT-B was pre-trained using the unsupervised masked image modeling technique and subsequently fine-tuned to be a classification model via supervised learning.

---

> > ### Comment · Reviewer_fXNW · 2023-08-18
> >
> > Thank you for your answer. I have to admit that it gave me a mixed impression: some points are left unanswered, and for others I did not find the answer satisfactory.
> >
> > ## Points left unanswered
> >
> > - I do not know how the 30+ techniques were selected. The methodology is missing.
> >
> > - I understand that the authors were unable to share the code base during the rebuttal. But currently, I cannot evaluate the claims of the paper about the code. Given that the main contribution of the paper is (as pointed by other reviewers) the software engineering work, I feel that I cannot evaluate properly this work as a whole.
> >
> > - Some limitations regarding the hyperparameter tuning are undressed. Cf below.
> >
> > ## Points not addressed satisfactorily
> >
> > - The explanations about the impossibility of the level of granularity are far from convincing, since the paper altered the taxonomy designed by  [59], and [59]  has a more granular level. The paper's taxonomy compares techniques that have different objectives (optimization and input augmentation, for example, or LGV/MoreBayesian and RFA) on the same basis (cf. my review). Directly re-using the taxonomy of [59] would strengthen the paper.
> >
> > - It is not enough to simply have non-overlapping sets of examples to tune hyperparameters. The sets of target models used to tune HPs should be distinct from the set of target models used to report the final results (cf. my review). Otherwise, the results overfit to some specific targets. This situation of data leakage corresponds to a threat model with query access to the target model. Since the paper positions it-self as a more realistic evaluation of transfer-based black-box attacks, the paper cannot be accepted with such flaw.
> >
> > - The rebuttal did not list specifically the techniques for which hyperparameters were selected through  cross-validation, in the current version of the paper.
> >
> > - In Table 1, most columns are empty for substitute model training. If tuning HPs is too computationally costly for LGV and MoreBayesian, simply report the results with the original HPs and a star indicating this. I believe that only the most important HPs could be tuned (for example, I doubt that tuning the weight decay is relevant for LGV).
> >
> > - If no random restart is performed (i.e. a single start from the  original image), then the attack must not be called PGD, but I-FGSM (or  equivalently BIM).
> >
> > - A fixed step size of I-FGSM independent of the perturbation norm is highly unlikely to be optimal. Please take into consideration my original remark.
> >
> > - More generally, I disagree that a benchmark should stay close to the experimental settings of current work, despite their addressable flaws and limitations. The goal of a good benchmark is to set a high standard of evaluation for past and future work.

---

> > > ### Comment · Area_Chair_gfE6 · 2023-08-18
> > >
> > > Thank the authors for the rebuttal. PCs and I have reminded the reviewers to respond to the rebuttals as soon as possible. The final decision will depend on both the reviews and rebuttal.
> > >
> > > @Reviewer fXNW: This message is yet another reminder. Please try to respond to the rebuttal asap.
> > >
> > > --AC

---

> > > ### Author Response · Authors · 2023-08-21
> > > **Response to the reviewer (part 1/5)**
> > >
> > > We feel sorry that our initial response in the rebuttal fails to convince the reviewer in some points. Since the rebuttal letter is limited to 6000 characters only, we had to compress our response hardly and thus some explanations may seem obscure or incomplete. We would like to thank the reviewer for pointing out comments that require further clarification and response. Please see our further clarification and responses as follows.
> > >
> > > &nbsp;
> > >
> > > > I do not know how the 30+ techniques were selected. The methodology is missing.
> > >
> > > **A:** We aim to test all state-of-the-art transfer-based methods for generating adversarial methods (that could compromise a general image classification victim model) published in top tier ML/CV conferences and journals, including NeurIPS, ICML, ICLR, CVPR, ICCV, ECCV, TPAMI, TIP, etc. We are very glad to add the missing methods which have been mentioned by the reviewer. If there exist other recent methods that can be compared, we are also more than glad to incorporate them into the benchmark.
> > >
> > > &nbsp;
> > >
> > > > But currently, I cannot evaluate the claims of the paper about the code.
> > >
> > > **A:** We hope to address the concerns of reviewers about the code as much as possible. In our general response, we have demonstrated how to evaluate a new method and how to register a new victim model for evaluation using our codebase. We would like to try our best to demonstrate them if there are any sections of the code that are specifically concerned.
> > >
> > > &nbsp;
> > >
> > > > Some limitations regarding the hyperparameter tuning are undressed.
> > > The sets of target models used to tune HPs should be distinct from the set of target models used to report the final results (cf. my review).
> > >
> > > **A:** Thanks for further clarifying your concern. We agree that it is important to report the final performance on some victim models which are distinct from those used on the validation set. We collected 15 additional victim models, including a BEiT-L, an EfficientNet-L2, a DeiT-L, a ConvNeXt V2-L, a Swin V2-L, a ViT-L, a CAFormer-B36, a MaxViT-L, an EVA-L, an EVA02-L, a MobileNet V2, a DenseNet-161, a ResNeXt-101, a SENet-154, and a RepVGG-B3, and conducted an experiment on attacking these victim models. For the "substitute model training" methods, the conclusion remains consistent with the observations from Table 1. Specifically, when employing I-FGSM as the back-end, RFA achieves the best AA (i.e., 63.98%), and when applying UN-DP-DI$^2$-TI-PI-FGSM as the back-end, MoreBayesian attains the best AA (i.e., 47.13%).
> > >
> > > For the "gradient computation" methods, we show the results in the below tables (in part 2/5 and 3/5). When I-FGSM is applied as the optimization back-end, we observe that the conclusion aligns with the findings in Table 1 of the paper. NAA consistently outperforms other methods on most choices of the substitute model, achieving the lowest AAA (i.e., 79.27%). When introducing UN-DP-DI$^2$-TI-PI-FGSM as the optimization back-end, the top three lowest AAs are achieved using ConvNeXt-B, DeiT-B, and Swin-B as the substitute models, as in Table 1. The best AA is obtained by performing LinBP on the ConvNeXt-B substitute model (i.e., 30.81%, which stands as the second-best in Table 1 and is only 0.20% higher than the best AA), due to slight distribution shift of the tested victim models. It is also noteworthy that, for each substitute model, the attack method that yields the lowest AA almost always remains consistent in the below table and Table 1.
> > >
> > > We would like to further explain that we focused on attacking the same victim models as those on the validation set mainly to explore the optimal performance of each method and to gain insights into their optimal performance when different substitute/victim models are presented. To be more specific, considering that we tested with different pairs of substitute and victim models, then the choice of validation victim models (if they are different from the test victim models) will largely affect conclusions that could be drawn. For example, if we were able to employ a ViT-S in the set of validation victim models, then the final performance of attacking ViT-B in practice using any substitute model would likely be better. This makes it difficult to obtain any insights regarding which substitute model should be chosen to gain higher attack success rates in practice, and it will lead to doubt about whether transferring from vision transformers to convolutional networks is really easier than from the opposite direction. We would like to add experimental results in the above table to our paper and highlight these points to avoid misleading.

---

> > > ### Author Response · Authors · 2023-08-21
> > > **Response to the reviewer (part 2/5)**
> > >
> > > |                               |  ResNet-50 |   VGG-19   | Inception v3 | EfficientNet v2 | ConvNeXt-B |    ViT-B   |   DeiT-B   |   BEiT-B   |   Swin-B   |   Mixer-B  |     AAA    |
> > > |-------------------------------|:----------:|:----------:|:------------:|:---------------:|:----------:|:----------:|:----------:|:----------:|:----------:|:----------:|:----------:|
> > > | - Baseline                    |            |            |              |                 |            |            |            |            |            |            |            |
> > > |             I-FGSM            |   89.95%   |   91.13%   |    95.21%    |      96.37%     |   89.53%   |   93.20%   |   93.73%   |   92.68%   |   95.88%   |   95.88%   |   93.36%   |
> > > | - Gradient   Computation      |            |            |              |                 |            |            |            |            |            |            |            |
> > > |           TAP (2018)          |   84.03%   |   89.09%   |    93.36%    |      95.72%     |   92.93%   |   94.66%   |   95.38%   |   94.66%   |   96.54%   |   95.73%   |   93.21%   |
> > > |          NRDM (2018)          |   83.39%   |   85.61%   |    88.15%    |      97.59%     |   96.35%   |   96.55%   |   96.86%   |   96.77%   |   96.77%   |   93.63%   |   93.17%   |
> > > |           FDA (2019)          |   86.43%   |   93.09%   |    92.23%    |      98.69%     |   97.36%   |   97.51%   |   96.94%   |   97.69%   |   98.09%   |   97.93%   |   95.60%   |
> > > |           ILA (2019)          |   77.71%   |   76.04%   |    86.88%    |      91.58%     |   87.87%   |   83.89%   |   87.14%   |   83.68%   |   91.37%   |   90.37%   |   85.65%   |
> > > |           SGM (2020)          |   76.87%   |      -     |       -      |      85.64%     |   80.16%   |   90.84%   |   92.02%   |   89.59%   |   93.42%   |   93.90%   |      -     |
> > > |          ILA++ (2020)         |   75.47%   |   73.55%   |    91.85%    |      89.83%     |   86.53%   |   81.79%   |   86.69%   |   82.42%   |   90.04%   |   88.98%   |   84.71%   |
> > > |          LinBP (2020)         |   78.77%   |   85.98%   |    94.84%    |      98.02%     |   91.95%   |   94.27%   |   94.39%   |   94.99%   |   96.69%   |   97.14%   |   92.70%   |
> > > |          ConBP (2021)         |   76.61%   |   84.77%   |       -      |        -        |      -     |      -     |      -     |      -     |      -     |      -     |      -     |
> > > |           SE (2021)           |      -     |      -     |       -      |        -        |      -     |   93.74%   |   93.35%   |   93.24%   |      -     |   94.96%   |      -     |
> > > |           FIA (2021)          | **74.01%** | **72.69%** |    87.48%    |      90.45%     |   83.36%   |   81.52%   |   84.98%   |   85.15%   |   89.70%   |   86.07%   |   83.54%   |
> > > |           PNA (2022)          |      -     |      -     |       -      |        -        |      -     |   92.25%   |   92.03%   |   91.94%   |   95.27%   |      -     |      -     |
> > > |           NAA (2022)          |   74.50%   |   77.62%   |  **81.59%**  |    **86.86%**   | **73.31%** | **76.57%** | **78.87%** | **74.60%** | **84.58%** | **84.21%** | **79.27%** |

---

> > > ### Author Response · Authors · 2023-08-21
> > > **Response to the reviewer (part 3/5)**
> > >
> > > |                          |  ResNet-50 |   VGG-19   | Inception v3 | EfficientNet v2 | ConvNeXt-B |    ViT-B   |   DeiT-B   |   BEiT-B   |   Swin-B   |   Mixer-B  |     AAA    |
> > > |--------------------------|:----------:|:----------:|:------------:|:---------------:|:----------:|:----------:|:----------:|:----------:|:----------:|:----------:|:----------:|
> > > | - Baseline               |            |            |              |                 |            |            |            |            |            |            |            |
> > > |  UN-DP-DI$^2$-TI-PI-FGSM |   52.38%   |   57.09%   |    69.70%    |      55.51%     |   42.48%   |   42.13%   |   48.85%   |   42.27%   |   48.68%   |   67.43%   |   52.65%   |
> > > | - Gradient   Computation |            |            |              |                 |            |            |            |            |            |            |            |
> > > |        TAP (2018)        |   71.84%   |   59.21%   |    75.30%    |      71.15%     |   38.75%   |   56.97%   |   63.83%   |   47.73%   |   62.06%   |   74.42%   |   62.13%   |
> > > |        NRDM (2018)       |   60.78%   |   63.85%   |    77.14%    |      63.83%     |   51.53%   |   64.71%   |   74.27%   |   61.71%   |   82.46%   |   76.11%   |   67.64%   |
> > > |        FDA (2019)        |   54.96%   |   55.67%   |    69.87%    |      92.98%     |   77.22%   |   96.78%   |   90.49%   |   86.03%   |   88.45%   |   96.69%   |   80.91%   |
> > > |        ILA (2019)        |   53.28%   |   51.67%   |    66.62%    |      49.89%     |   41.39%   |   41.04%   |   50.83%   |   40.47%   |   63.99%   |   67.31%   | **52.65%** |
> > > |        SGM (2020)        | **49.87%** |      -     |       -      |      51.61%     |   33.54%   | **39.80%** |   40.91%   | **37.29%** | **31.88%** |   61.79%   |      -     |
> > > |       ILA++ (2020)       |   53.00%   | **51.24%** |  **66.27%**  |    **50.34%**   |   41.70%   |   41.19%   |   51.55%   |   40.41%   |   66.93%   |   66.97%   |   52.96%   |
> > > |       LinBP (2020)       |   52.97%   |   56.00%   |    85.62%    |      93.87%     | **30.81%** |   53.00%   |   49.74%   |   54.79%   |   74.03%   |   89.76%   |   64.06%   |
> > > |       ConBP (2021)       |   51.69%   |   56.04%   |       -      |        -        |      -     |      -     |      -     |      -     |      -     |      -     |      -     |
> > > |         SE (2021)        |      -     |      -     |       -      |        -        |      -     |   49.70%   | **36.42%** |   38.79%   |      -     | **61.26%** |      -     |
> > > |        FIA (2021)        |   53.63%   |   59.96%   |    69.64%    |      79.53%     |   57.92%   |   51.26%   |   53.32%   |   64.99%   |   64.75%   |   74.17%   |   62.92%   |
> > > |        PNA (2022)        |      -     |      -     |       -      |        -        |      -     |   44.36%   |   37.73%   |   41.19%   |   36.01%   |      -     |      -     |
> > > |        NAA (2022)        |   53.93%   |   57.44%   |    67.18%    |      57.69%     |   41.89%   |   43.77%   |   46.53%   |   49.65%   |   53.72%   |   63.04%   |   53.48%   |

---

> > > ### Author Response · Authors · 2023-08-21
> > > **Response to the reviewer (part 4/5)**
> > >
> > > > The explanations about the impossibility of the level of granularity are far from convincing, since the paper altered the taxonomy designed by [59], and [59]  has a more granular level. The paper's taxonomy compares techniques that have different objectives (optimization and input augmentation, for example, or LGV/MoreBayesian and RFA) on the same basis (cf. my review). Directly re-using the taxonomy of [59] would strengthen the paper.
> > >
> > > **A:** We appreciate your further comment, but we respectfully insist the taxonomy of our paper.
> > >
> > > In our paper, LGV and MoreBayesian are categorized into "substitute model training", together with RFA, since they all propose principled substitute model training/fine-tuning strategies and **require the attacker to own the training data** (which is very different from other methods). By contrast, together with LinBP and SGM, RFA is categorized into "surrogate refinement" methods in the independent work of [59]. We agree that, from a certain perspective, we can say that RFA, LGV, and MoreBayesian also modify the substitute/surrogate models, but we believe that the important difference between their threat models (about access to the training resource and training data) should not be ignored in the taxonomy. Additionally, we deem that separating these methods from "surrogate refinement" methods also provide a more granular taxonomy for these methods.
> > >
> > > For methods in the category of input augmentation and optimizer, they share many commonalities, making it reasonable to study them together. To be more specific, they are all inspired by model training techniques that prevent models from getting stuck in local minima, and they are not specific to substitute model architectures and do not require training data, unlike the other methods. In fact, rather than comparing input augmentation methods and the optimizers with each other in isolation, we put them in combination to test their effectiveness, as in [43, 7], considering that it is natural to adopt a combination of these methods (just like in training DNN models). We believe this can be a more reasonable way of testifying methods in this category and we would like to advocate it to future work in the adversarial machine learning community.
> > >
> > > &nbsp;
> > >
> > > > The rebuttal did not list specifically the techniques for which hyperparameters were selected through cross-validation, in the current version of the paper.
> > >
> > > **A:** We tuned architecture-related hyper-parameters for transfer-based attacks, including the choice of position for NRDM, ILA, ILA++, LinBP, ConBP, FIA, and NAA and the scaling factor for SGM, since we tested with a variety of substitute architectures and the suggested hyper-parameters in these papers may not be suitable to a different substitute architecture than the ones tested in these papers. We compare the performance of all possible values of these hyper-parameters and chose the best ones on the validation set. We will also include in the paper the specific values of these hyperparameters for each substitute model.
> > >
> > > &nbsp;
> > >
> > > > In Table 1, most columns are empty for substitute model training. If tuning HPs is too computationally costly for LGV and MoreBayesian, simply report the results with the original HPs and a star indicating this. I believe that only the most important HPs could be tuned (for example, I doubt that tuning the weight decay is relevant for LGV).
> > >
> > > **A:** We appreciate the suggestion about tuning only the most important hyper-parameters. We are glad to conduct such an experiment. The experiment is ongoing and the results will be added to the revised paper. Note that, as reported by previous work, the training of some tested models requires substantial effort to reach convergence, especially with a modified training objective.

---

> > > ### Author Response · Authors · 2023-08-21
> > > **Response to the reviewer (part 5/5)**
> > >
> > > > If no random restart is performed (i.e. a single start from the original image), then the attack must not be called PGD, but I-FGSM (or equivalently BIM).
> > >
> > > **A:** The main difference in the design between PGD and I-FGSM is that PGD initializes the perturbation with a random tensor sampled from a distribution, while I-FGSM initializes the perturbation with a zero tensor as discussed in lines 101-103 in the paper. We would like to highlight in the paper that, without multiple restarts, the difference between these two methods lies only in the initialization, and this is the reason why the performance of I-FGSM and PGD is similar.
> > >
> > > &nbsp;
> > >
> > > > A fixed step size of I-FGSM independent of the perturbation norm is highly unlikely to be optimal. Please take into consideration my original remark.
> > >
> > > **A:** We really appreciate the suggestion regarding the exploration of more advanced optimizer with an adaptive step size. Although we followed the setting of previous works in experiments with I-FGSM, we are open to adding additional evaluation as per your suggestion.
> > >
> > > &nbsp;
> > >
> > > > More generally, I disagree that a benchmark should stay close to the experimental settings of current work, despite their addressable flaws and limitations. The goal of a good benchmark is to set a high standard of evaluation for past and future work.
> > >
> > > **A:** We fully agree that a high standard of evaluation should be expected. Previous work may have unaddressed flaws and limitations, as you have pointed out, and we have actually tried to demonstrate and address some most critical ones (from our perspective) in this paper, e.g., the pre-processing operations as discussed in lines 194-215, the optimization back-end as discussed in lines 277-282, and the substitute architectures. In the meanwhile, our experiments still inherit some settings from previous work, and we would like to explain that this is designed mainly to ablate other affecting factors when demonstrating those most critical points in the first place. We aim to keep addressing more problems and deliver more messages about the standard of evaluation. Some initial results have been given in the rebuttal and we will add them to our paper, e.g., the results on defensive victim models and the results on victim models different from those have been evaluated on the validation set.

---

### Official Review · Reviewer_xpVS · 2023-07-09

**Soundness:** 3 good
**Presentation:** 3 good
**Contribution:** 2 fair
**Rating:** 6
**Confidence:** 3

**Summary:**

This paper is a benchmark paper on transfer-based attack in the area of adversarial machine learning. The paper benchmarks 30+ methods on ImageNet, grouped into four principal categories: augmentation and optimizer; gradient computation; substitute model training, and generative models. The extensive evaluation results uncover several interesting novel insights. For instance, employing Vision Transformer (ViT) architectures in the training of substitute models can generally enhance the efficacy of transfer attacks. Additionally, the MoreBayesia method has consistently demonstrated an improved transferability of adversarial examples.

**Strengths:**

1.	The evaluation is comprehensive, encompassing over 30 implemented and evaluated methods, thereby offering a thorough comparison of state-of-the-art transfer-based attacks.
2.	Additionally, this benchmark investigates the robustness of transfer-based attacks on ViT, an aspect often overlooked in other papers.
3. The evaluation results provide novel insights into the design of more potent transfer-based attacks.


**Weaknesses:**

The paper conducts an extensive and systematic study on transfer-based attacks, comparing state-of-the-art approaches. However, one concern is the absence of significant novel insights. While the paper does offer interesting findings, such as the improved performance of transfer attacks when using ViT as substitute models, it could benefit from additional unique insights to meet the standards expected at NeurIPS.


Furthermore, the paper lacks a discussion on other types of black-box attacks, despite transfer-based attacks being a specific form of such attacks. It would be valuable to include a discussion on related work concerning different black-box attack techniques, highlighting the similarities and differences between various approaches. This would provide a more comprehensive understanding of transfer-based attacks in the broader context of black-box adversarial attacks.


**Questions:**

See the weakness part.

**Limitations:**

Yes.

---

> ### Author Rebuttal · Authors · 2023-08-09
>
> Thanks for the positive feedback. Our responses to the comments are given as follows.
> &nbsp;
>
> > One concern is the absence of significant novel insights.
>
> **A:** The novelty of our paper shows in several different aspects. First of all, our work develops a new combination of input augmentation and optimizer techniques, which surprisingly outperforms all other sorts of methods, and it can be considered as a strong and useful optimization back-end for developing new transfer-based attacks. Moreover, as recognized by many other reviewers, our paper delivers many novel results and insights, including 1) it is easier to transfer from vision transformers to convolutional networks than from the opposite direction, 2) it is essential to evaluate on a variety of substitute and victim models to gain a comprehensive understanding of the performance of a transfer-based method, _etc_. These insights are expected to contribute to the development of future work in the field of trustworthy machine learning and inspire effective adversarial machine learning methods.
> &nbsp;
>
> > Despite transfer-based attacks being a specific form of such attacks, it would be valuable to include a discussion on related work concerning different black-box attack techniques.
>
> **A:** Thanks for the suggestion. The suggested discussions about attacks other than transfer-based ones will be added to the paper.

---

> > ### Comment · Reviewer_xpVS · 2023-08-16
> > **Thanks for the rebuttal**
> >
> > Thank the authors for the rebuttal. After going through the author's rebuttal and other reviews, some of my concerns are addressed, so I decided to increase my rating to weak accept. The reason that I don’t further increase my rating is that more interesting findings or insights can be expected.

---

> > > ### Author Response · Authors · 2023-08-17
> > > **Thanks to the reviewer**
> > >
> > > Dear Reviewer xpVS,
> > >
> > > It's excellent to know that your concerns have been addressed! We value your suggestion and will delve into more interesting findings and insights in future work.
> > >
> > > Best regards,
> > > Authors

---

### Official Review · Reviewer_EHYy · 2023-07-16

**Soundness:** 3 good
**Presentation:** 4 excellent
**Contribution:** 3 good
**Rating:** 7
**Confidence:** 4

**Summary:**

In this paper, the authors present a new transfer-based attack benchmark (TA-Bench) to evaluate the transferability of adversarial attacks. TA-Bench implements 30+ adversarial attacks with 10 substitute models and introduces more advanced optimization back-ends that incorporate augmentation and different choices of optimizers. The benchmark provides a means to compare different adversarial attacks systematically, fairly and practically. Evaluation results bring new and interesting insights about existing attacks.

**Strengths:**

1. This is a solid paper with clear motivation and comprehensive experimental results.
2. The proposed TA-Benchmark provides a systematical, practical and fair way to compare different adversarial attacks.
3. TA-benchmark covers attacks with various mechanisms,  including the implementation of many more recent and advanced attacks.
4. The authors bring new insights about existing adversarial attacks such as choices of augmentation and optimizer have impacts on effectiveness and transformers are better substitute models for gradient computation-based attacks.
5. The paper is well written and easy to follow. I enjoyed reading this paper.

**Weaknesses:**

1. The codebase is not provided (maybe for anonymity). Thus it is hard to evaluate the accessibility, portability, scalability and usability of the benchmark. For example, how difficult would it to evaluate a new attack using TA-bench? Is there any interface for users to quickly implement their own attacks or select their own substitute models?
2. Since the main contribution of this work is providing a fair and practical benchmark to evaluate different attacks. It is worth discussing the standards and metrics for the evaluation. However, I don't find an explicit explanation of such standards and metrics.
3. The adversarial perturbations ($\epsilon$) are set to be fixed in the experiments. However, attackers may use different $\epsilon$ in practice.

**Questions:**

Please see weaknesses for details.

**Limitations:**

No potential negative societal impact.

---

> ### Author Rebuttal · Authors · 2023-08-09
>
> Thanks for the positive feedback. Except for the question about codebase which is answered in our general response, all comments are replied to as follows.
> &nbsp;
>
> > It is worth discussing the standards and metrics for the evaluation. However, I don't find an explicit explanation of such standards and metrics.
>
> **A:** The metrics and standards are introduced in lines 167-176 in our paper. To assess the transferability of adversarial examples, we first evaluated the prediction accuracy of all victim models given adversarial examples generated on a substitute model. Based on the prediction accuracy of each victim model, we further calculate the average accuracy (AA) of all models. Moreover, the benchmark also calculates average AA (AAA), worst AA (WAA), and best AA (BAA) on all choices of substitute models, and they are reported for almost all experiments in our paper. Taking the prediction accuracy instead of the success rates as a base metric makes it possible to add more substitute models in the future, as calculating the success rates in principle requires images that could be correctly classified by all substitute models to generate adversarial examples.
> &nbsp;
>
> > The adversarial perturbations ($\epsilon$) are set to be fixed in the experiments. However, attackers may use different $\epsilon$ in practice.
>
> **A:** We follow the setting of using $\epsilon=8/255$ for $\ell_\infty$ attacks and $\epsilon=5$ for $\ell_2$ attacks, as in previous papers. We will consider performing additional evaluations using other $\epsilon$ values if they are commonly adopted.

---

> > ### Comment · Reviewer_EHYy · 2023-08-10
> >
> > I have read the rebuttal. Given the strengths of the paper, I tend to keep my rating. However, I still want to see the performance against different perturbations (e.g, $\epsilon$ = 16/255 for $l_{\infty}$ and 2 for $l_2$).

---

> > > ### Author Response · Authors · 2023-08-11
> > >
> > > Dear Reviewer EHYy,
> > >
> > > Thanks for recognizing the strengths of our paper. Experiments on the suggested $\epsilon$ values are currently ongoing. We will provide some preliminary results as soon as they are available, which will likely be in several days.
> > >
> > > Best regards,
> > > Authors

---

> > > ### Author Response · Authors · 2023-08-14
> > >
> > > Dear Reviewer EHYy,
> > >
> > > We have obtained the results under the $\ell_\infty$ constraint with $\epsilon$=16/255, using our new optimization back-end. The conclusions are similar to those from the lower half of Table 1 in our paper. The best attack performance in the sense of BAA is still achieved by applying SE on the DeiT-B substitute model, resulting in the victim models showing an average accuracy of only 4.36%. PNA leads to the lowest WAA among all, which is 10.34%. For the "substitute model training" methods, MoreBayesian still outperforms the other methods, successfully fooling the victim models to show an average accuracy of only 6.59% when using a ResNet-50 substitute model. More detailed results will be added to our paper, and the results for $\ell_2$ attacks with $\epsilon$=2 will also be added.
> > >
> > > Best regards,
> > > Authors

---

> > > > ### Comment · Reviewer_EHYy · 2023-08-15
> > > >
> > > > Thank you for the updated results. Looks all good to me. I'd like to keep my rating as Accept.

---

### Official Review · Reviewer_urLG · 2023-07-20

**Soundness:** 3 good
**Presentation:** 3 good
**Contribution:** 3 good
**Rating:** 6
**Confidence:** 3

**Summary:**

In this paper, the authors have presented benchmark for transfer-based attacks, in which they have implemented 30+ advanced transfer-based attack methods, including those focus on augmentation and optimizer innovation, those “gradient computation” methods, those “substitute model training” methods, and those applying generative modeling.  And by evaluating and comparing transfer-based attacks systematically, they have some new insights.

**Strengths:**

1. This paper is the first transfer-based attack benchmark, it will help people in this field.

2. The authors have implemented 30+ methods, including augmentation, optimizer, gradient computation, substitute model training, and generative model. Various methods make sure its fairness and comprehensiveness.

3. It is helpful that the results of the experiment show some beneficial conclusions.


**Weaknesses:**

1. There is only one type of dataset used in the experiment, which makes results lack credibility.

2. This paper mainly evaluates transfer-based methods on the base of two back-ends, and the results are quite different. It lacks more explanation.

3. The authors have maked a bilinear interpolation and resize in the pre-processing, which has an influence on following evaluations.


**Questions:**

1. Why is attack performance of ILA, ILA++, and NAA using I-FGSM quite different with that using new back-end?

2. The analysis of Figure 3 in Section 4.3 is too short to make readers understand why the most effective factor of these methods may be input augmentation and gradient averaging.


**Limitations:**

See Weaknesses.

---

> ### Author Rebuttal · Authors · 2023-08-09
>
> Thanks for the positive feedback. Our responses to the comments are given as follows.
> &nbsp;
>
> > There is only one type of dataset used in the experiment, which makes results lack credibility.
>
> **A:** We focus on ImageNet first for several reasons. First of all, almost all papers studying transfer-based attacks **developed** and **evaluated** their methods on ImageNet. Only a few papers conducted evaluations on smaller scale datasets, such as CIFAR-10, as well. This is because ImageNet contains large-scale diverse images, making the data distribution more representative. We ensure a fair comparison on ImageNet first to benefit understanding what's effective. Another reason why we focus on ImageNet first is that vision transformers generally require training on ImageNet, and the dataset offers more options for choosing substitute/victim models, enabling a more comprehensive understanding of the performance of an attack, as discussed in Section 4.3. After consolidating all results on ImageNet, we will consider performing evaluations CIFAR-10, too.
> &nbsp;
>
> > This paper mainly evaluates transfer-based methods on the base of two back-ends, and the results are quite different. It lacks more explanation. Why is attack performance of ILA, ILA++, and NAA using I-FGSM quite different with that using new back-end?
>
> **A:** The advanced optimization back-end (_i.e._, UN-DP-DI$^2$-TI-PI-FGSM) is developed by combining effective methods in the category of "augmentation and the optimizer." Different results achieved using UN-DP-DI$^2$-TI-PI-FGSM, compared to the results on the typical I-FGSM back-end, come from the fact that, in certain methods, the key points for improving the transferability overlap with those in the methods used in UN-DP-DI$^2$-TI-PI-FGSM. For example, NAA achieves the best AAA with the I-FGSM back-end, while it fails to maintain its edge when UN-DP-DI$^2$-TI-PI-FGSM is adopted. A very recent paper [1] has analyzed NAA from the gradient alignment perspective, and it claims that NAA resembles changing the intermediate-level features of the adversarial example into that of some randomly augmented benign examples. This can be similarly achieved by random input transformation methods, leading to less effective performance when combining them together. The same reason may also lead to unsatisfactory performance of some other methods. More analyses will be given in an updated version of the paper.
> &nbsp;
>
> > The authors have maked a bilinear interpolation and resize in the pre-processing, which has an influence on following evaluations.
>
> **A:** We would like to stress that the bilinear interpolation and resize operations mentioned in lines 161-164 are only for the ResNet-50 victim, and it is actually the official implementation of the model. We strictly adhere to the official pre-processing pipeline for each model (as mentioned in lines 159-161).
> &nbsp;
>
> > The analysis of Figure 3 in Section 4.3 is too short to make readers understand why the most effective factor of these methods may be input augmentation and gradient averaging.
>
> **A:** We would like to provide more analyses in the paper. In general, since these methods all apply random augmentation and gradient averaging, and there lacks comparison to a baseline that uses the same random augmentation and gradient averaging strategies, we conducted experiments to ensure such a fair comparison (Figure 3). The inferior performance of these methods compared to the baselines indicates that the random augmentation and gradient averaging approach contributes the most.
> &nbsp;
> &nbsp;
>
> [1] Qizhang Li, et al. Improving Adversarial Transferability by Intermediate-level Perturbation Decay. In arXiv 2023.

---

> > ### Comment · Reviewer_urLG · 2023-08-14
> > **Response to the rebuttal**
> >
> > Dear authors,
> >
> > Thanks for your response. Your response addresses most of my concerns. Thus, I am inclined to give weak accept.

---

> > > ### Author Response · Authors · 2023-08-14
> > > **Thanks to the reviewer**
> > >
> > > We would like to thank the reviewer for responding to our rebuttal. It's great to know that most of your concerns have been addressed!

---

### Author Rebuttal · Authors · 2023-08-09

We would like to thank all reviewers for the valuable feedback. Our responses to some common questions are given as follows.
&nbsp;

> The codebase.

**A:** As promised in the paper, the codebase will be made publicly available. With the codebase, APIs are directly provided for evaluating attacks using substitute/victim models in the paper. Users could also register their own substitute models. Unfortunately, it is not allowed to upload supplementary material during the rebuttal period (links to external pages are also not permitted), and we here try to provide some code snippets of our TA-Bench. For instance, the way of evaluating the transferability of adversarial examples crafted by any new method:
  ```
  import tabench
  evaluator = tabench.Evaluation(data_dir="/path/to/adv/examples", mode="standard")
  evaluator.evaluate()
  ```
  If one aims to register a new model as the victim model, then the implementation can be simply formatted as follows.
  ```
  import tabench

  ##### For victim models that exist in timm
  victims=["deit3_huge_patch14_224", "vit_small_patch16_224"]
  evaluator = tabench.Evaluation(data_dir="/path/to/adv/examples", mode="custom-timm", victims=victims)
  evaluator.evaluate()

  ##### For victim models that do NOT exist in timm
  victims = [{"model_name": "your_preferred_model", "model": model, "preprocessing": transforms},]
  evaluator = tabench.Evaluation(data_dir="/path/to/adv/examples", mode="custom-custom", victims=victims)
  evaluator.evaluate()
  ```
&nbsp;

> A perfect fit for the B&D track.

**A:** We appreciate the recognition of the value of this work. Our work develops a new combination of input augmentation and optimizer techniques, which outperforms all other sorts of methods and can be considered as a strong optimization back-end for developing new transfer-based attacks. Moreover, as recognized by most reviewers, our paper delivers many novel results and insights, including 1) it is easier to transfer from vision transformers to convolutional networks than from the opposite direction, 2) it is essential to evaluate on a variety of substitute and victim models to gain a comprehensive understanding of the performance of a transfer-based method, _etc_. These insights are also expected to contribute to the development of future work in the field of adversarial machine learning and inspire effective methods for generating adversarial examples to evaluate the robustness of DNNs. Thus, we consider it suitable not only for the benchmark and datasets track but also for the audience of the main track.
&nbsp;

> Taxonomy and categories of methods.

**A:** We categorize transfer-based attacks based on their commonalities.
* **Input augmentation and the optimizer.** Inspired by the evaluation of some newly developed computer vision architectures (as discussed in lines 217-226), we put "input augmentation" and "optimizer" methods together to evaluate how empirical evaluation of the other sorts of methods can be biased with less optimal augmentation and optimizers. These methods are generally not specific to substitute model architectures and also do not require training data, unlike the other methods.
* **Gradient computation.** The methods in this category all attempt to improve the transferability of adversarial examples by modifying the loss or the backpropagation process.
* **Substitute model training.** The methods in this category all developed principled substitute model training/fine-tuning strategies.
* **Generative Modeling.** These methods advocate training a generative model first and then generating adversarial examples.

We feel that it is challenging to further divide these methods into more granular categories. Most of the methods that are "feature attack" (suggested by *Reviewer Yj8S*) can also be considered as discarding higher layers of the substitute model (_e.g._, NRDM, ILA, and ILA++), thus can also be considered to be related to the "network structure" category (suggested by *Reviewer Yj8S*). Methods like LinBP not only modify architectures of the substitute models but also operate only after a middle layer, which can be seen to modify the gradient with respect to the "features" and is also related to "feature attack". We have talked to the authors of some of these works about the taxonomy and found it difficult to divide them into granular categories.
&nbsp;

> Defense methods and targeted attacks.

**A:** Although it can be insightful to include defensive models in the benchmark, considering that in our experiments each victim model is also testified as a substitute model (the necessity of such a setting is discussed in Section 4.3), the computational complexity scales quadratically with the number of models, thus we only include some popular models in practice in the first version. After submission, we tried evaluating the performance of different methods in attacking 3 defensive models obtained via adversarial training, _i.e._, a robust ConvNext-B, a robust Swin-B, and a robust ViT-B-CvSt. They are all collected from RobustBench [1] and exhibit high robust accuracy against AutoAttack. The results are given in the attached PDF and will be added to our paper.

As for targeted attacks, unfortunately, **most existing transfer-based attacks** were still developed in an untargeted setting. It is nontrivial to adapt them to the targeted setting and ensure their optimality. Thus, in order to compare these methods fairly in the first place, we think it is more reasonable to stick with the same untargeted setting in the first version of our benchmark. Also, as mentioned by *Reviewer Yj8S*, the targeted attack is more like a future direction, and we will test it, once appropriate.

We will discuss these points in the paper following the suggestion from *Reviewer fXNW*.
&nbsp;
&nbsp;

[1] Francesco Croce et al. RobustBench: a standardized adversarial robustness benchmark. In arXiv 2020.

---

### Decision · Program_Chairs · 2023-09-21

**Decision:**

Accept (poster)

**Comment:**

Seven experts reviewed the paper, and all liked the paper's benchmark for transfer-based attacks, extensive experiments, and some interesting findings. Hence, AC recommends the paper for acceptance.

AC hopes the authors will release the code as soon as possible since multiple reviewers have pointed out that it is an essential part of the benchmark. Besides, AC encourages the authors to incorporate the rebuttals and the reviewers' comments into the revised paper.

Reviewer Yj8S greenlighted the paper in a private communication with AC, and yet suggested the benchmark "reflects the focus on defense methods and targeted attacks."